Resource

# Deep single-cell decoding of human pancreatic islets reveals T2D β-cell gene expression defects

Khushdeep Bandesh [ID][1,6], Efthymios Motakis[1,6], Siddhi Nargund [ID][1,6], Romy Kursawe [ID][1], Vijay Selvam[1], Ansarullah [ID][2], Redwan M Bhuiyan[1,3], Giray Naim Eryilmaz [ID][1], Amelia M Willett [ID][2], Jacqueline K White[2], Sai Nivedita Krishnan[1,3], Cassandra N Spracklen [ID][4], Duygu Ucar[1,3,5] & Michael L Stitzel [ID][1,3,5 ✉]

## Abstract

Pancreatic islets maintain glucose homeostasis through coordinated action of endocrine and affiliate cell types and are central to type 2 diabetes (T2D) genetics and pathophysiology. Our understanding of robust human islet cell type-specific alterations in T2D remains limited. Here, we report comprehensive single-cell transcriptome profiling of 245,878 human islet cells from 48 donors spanning non-diabetic, pre-diabetic, and T2D states, and we identify 14 distinct cell types detected in every donor. Cell-cluster analysis reveals ~25–30% β-cell reductions consisting of β-cell loss and proportional increases in a senescent β-cell subpopulation in T2D donors, consistent with previous reports. Further, comparative data integration identifies 511 differentially expressed genes (DEGs) in T2D β-cells, including T2D-associated vitamin A metabolism genes, which are linked to impaired β-cell viability by multimodal functional validation. Integration with T2D genetic, pro-teomic, and mouse model metabolic phenotypes nominates 58 candidate causal T2D genes, including *PDZK1* and *GRAMD2B*, which preserve β-cell mass. Together, this genomic resource provides an enhanced type 2 diabetes expression-atlas for data exploration, analysis, and hypothesis testing, as well as a novel genomic resource for insights into T2D pathophysiology and human islet dysfunction.

**Keywords** Human Islet scRNA-seq; Type 2 Diabetes (T2D); Beta-cell Death; Vitamin A Metabolism; *GRAMD2B*
**Subject Categories** Metabolism; Molecular Biology of Disease

## Introduction

Type 2 diabetes afflicts >500 million people worldwide (International Diabetes Federation, https://diabetesatlas.org). Pancreatic islets are central mediators of type 2 diabetes (T2D) genetic risk and pathophysiology, driving insulin secretion defects (Thurner et al, 2018; Stitzel et al, 2010; Varshney et al, 2017). They comprise multiple cell types, including insulin-secreting beta (β) cells, glucagon-secreting alpha (α) cells, somatostatin-secreting delta (δ) cells, pancreatic polypeptide-secreting gamma (γ) cells, and ghrelin-secreting epsilon (ε) cells, that collectively determine islet functional output (Cabrera et al, 2006). In humans, they are intermingled throughout the islet, ensuring equal access to vasculature to sense and respond to fluctuating glucose levels (Noguchi and Huising, 2019). Growing evidence shows that α- and δ-cell signals regulate β-cell function to ensure proper insulin secretion dynamics (Noguchi and Huising, 2019). Robust assessment of islet cell type-specific gene expression programs and their regulation in pathologic states is crucial to define and understand pancreatic dysfunction in T2D. However, donor variability, modest sample size, and/or a relatively small number of cells sampled per individual have limited power to detect robust, reproducible differences (Elgamal et al, 2023; Fang et al, 2019; Weng et al, 2023; Segerstolpe et al, 2016; Lawlor et al, 2017).

To address this challenge, we completed single-cell transcriptome profiling and analysis of pancreatic islets obtained from a cohort of 48 non-diabetic (ND), pre-diabetic (PD), and type 2 diabetic (T2D) donors. We identified robust T2D-associated differences in islet β-cell proportions and gene expression. Integration of β-cell differentially expressed genes (DEGs) with complementary experimental, physiologic, and genetic data and analyses nominated 58 candidate T2D causal genes, and we functionally validated 12 as modulators of β-cell viability. These include *PDZK1* and *GRAMD2B*, two downregulated T2D β-cell genes whose inhibition increased islet β-cell death and impaired glucose homeostasis, establishing them as causal genes contributing to islet dysfunction and death.

## Results

### Comprehensive human islet single-cell transcriptome atlas spanning non-diabetic, pre-diabetic, and type 2 diabetic states

To build a comprehensive, representative atlas of human islet transcriptomes, we completed single-cell RNA sequencing

[1]The Jackson Laboratory for Genomic Medicine, Farmington, CT, USA. [2]The Jackson Laboratory, Center for Biometric Analysis (CBA), Bar Harbor, ME, USA. [3]Department of Genetics and Genome Sciences, UConn Health, Farmington, CT, USA. [4]Department of Biostatistics and Epidemiology, University of Massachusetts Amherst, Amherst, MA, USA. [5]Institute for Systems Genomics, UConn, Farmington, CT, USA. [6]These authors contributed equally as first authors: Khushdeep Bandesh, Efthymios Motakis, Siddhi Nargund. ✉E-mail: michael.stitzel@jax.org

(scRNA-seq) in human pancreatic islets obtained through the Integrated Islet Distribution Program (IIDP) from a diverse cohort of 48 American cadaveric organ donors, representing European, Hispanic, and African American self-reported ancestries and independent from those included in the Human Pancreas Analysis Program (HPAP) (Elgamal et al, 2023; Patil et al, 2023) (Table EV1). The cohort included 17 diagnosed T2D (mean HbA1c = 7.6%), 14 PD (mean HbA1c = 5.9%; designated based on American Diabetes Association (ADA) prediabetes criteria (5.7% ≤ HbA1c ≤ 6.4%) (American Diabetes Association Professional Practice Committee, 2021)), and 17 ND (mean HbA1c = 5.2%) donors (Fig. 1A). ND control donor samples were selected to be as similar as possible to the T2D cases with respect to age, sex, BMI, and self-reported ancestry ("Methods"). HbA1c levels differed significantly between groups, but age and BMI were similar (Fig. 1B, Games–Howell post-hoc test). Sex and ancestry distributions were comparable (Table EV1). Donor islets were dissociated into single-cell suspensions, which were captured and sequenced to a median sequencing depth of 13,400 UMIs using droplet-based scRNA-seq (10X Genomics). In total, 245,878 cells passed stringent quality control ("Methods"), with an average of 5122 high-quality single cells per donor (Appendix Fig. S1a,b; Dataset EV1). This cohort yielded more high-quality cells per donor (1.73×, Appendix Fig. S2a), a higher proportion of β-cells (3.61×) and other endocrine cell types, and fewer contaminating acinar cells (0.37×) than a similarly sized cohort (Elgamal et al, 2023). In addition, deeper sequencing per cell type (Appendix Fig. S2b) delivered more expressed genes detected per cell type except proliferating α-cells (Appendix Fig. S2c). After batch correction (Appendix Fig. S1c), unsupervised clustering based on expression of the 2000 most variable genes among these single-cell transcriptomes identified 14 distinct cell types corresponding to endocrine (α, proliferating α, β, δ, γ, and ε), exocrine (acinar and ductal), stellate/quiescent stellate, endothelial, glial (Schwann), and resident (mast) and infiltrating immune cell types across ND, PD, and T2D donors (Fig. 1C; Appendix Fig. S3a–c).

We defined robust signature genes—those specific to each islet cell type and detected across donors irrespective of their glycemic status—by aggregating and comparing per-donor single-cell transcriptomes of individual cell types. In addition to classic hormone-encoding marker genes such as *INS*, *GCG*, *SST*, *PPY*, and *GHRL*, we identified 270 α-, 272 β-, 173 δ-, 130 γ-, and 194 ε-cell signature genes exhibiting ≥ 8-fold expression differences at a false discovery rate (FDR) < 5% as estimated by Seurat LR model in one vs all comparison (Dataset EV2). Functional processes enriched among these signature genes included G protein-coupled receptor signaling and amino acid transport (α-cells); insulin secretion, regulation of membrane potential, and neuronal transmission (β-cells); gamma-aminobutyric acid signaling/synaptic transmission and synapse assembly (δ-cells); G protein-coupled receptor signaling, neuropeptide signaling, and regulation of cation channel activity (γ-cells); and regulation of lipoprotein lipase activity (ε-cells) (Dataset EV3). Examining sex differences, we compared gene expression between males and females across all states combined, identifying 112 α-, 64 β-, and 45 δ-cell DEGs by sex, 27 of which were shared across these three cell types (Dataset EV4). In all, 26/27 sex-specific DEGs were on X or Y chromosomes and were not significantly enriched for any biological processes or pathways.

## Significant β-cell loss and subpopulation changes in T2D donor islets

After establishing the islet cell types and their robust expression signatures, we assessed T2D-associated alterations in islet cell type composition by comparing cell type counts obtained from each islet donor between ND, PD, and T2D states (Dataset EV5). The distribution of each cell type relative to the total number of cells confirmed substantial inter-donor heterogeneity within each state (Fig. 1D) (Fang et al, 2019; Weng et al, 2023; Wang et al, 2023). Consistent with previous reports (Wu et al, 2021; Wang et al, 2023), overall β-cell/endocrine proportions were 13-15% lower in T2D islets compared to ND or PD donor islets (Fig. 1E; mean β-cell percentages: T2D = 42.2 ± 11.3; ND = 55.2 ± 10.7, *P* = 0.006; PD = 57.2 ± 12.9, *P* = 0.002, ANOVA followed by Tukey's honest significance test). α-cell proportions were correspondingly higher in T2D islets (48.7 ± 11.2% vs. 35.8 ± 10.5% in ND (*P* = 0.006) or vs. 35.7 ± 13.6% in PD (*P* = 0.009)). Relative δ- and γ-cell proportions were comparable between groups. Collective analysis of the full cohort revealed a significant inverse correlation between reduced β-cell proportions and elevated HbA1c levels (Fig. 1F, Spearman's *r* = −0.39; *P* = 0.006), consistent with reported inverse associations between HOMA-β (an index reflecting β-cell function) and HbA1c levels (Hou et al, 2016), while α-cell expansion also correlated with elevated HbA1c (Fig. 1G, Spearman's *r* = 0.38; *P* = 0.007).

Islets from 30/48 donors in this single-cell transcriptome cohort were also characterized by the IIDP Human Islet Phenotyping Program (HIPP) (https://iidp.coh.org/Resources-Offered/HIPP), which reported immunofluorescence-based estimates of their islet cell type composition (Dataset EV5). Cell proportions calculated from per-donor scRNA-seq profiles and HIPP for these samples correlated (Appendix Fig. S4; ND *r* = 0.62, PD *r* = 0.70, and T2D *r* = 0.73 for β-cells; ND *r* = 0.65, PD *r* = 0.81, and T2D *r* = 0.74 for α-cells), suggesting that the scRNA-seq-determined cell proportion differences were not due to cell loss during sample processing or single-cell capture.

Single-cell and targeted islet analyses have reinvigorated interest in and debates around models in which (patho)physiologic states such as T2D are characterized by molecularly and functionally distinct endocrine cell subpopulations or states with variable maturity, stress, hormone secretion and glucose responsiveness (Campbell et al, 2020; Carril Pardo et al, 2022; Wang et al, 2024; Fu et al, 2023; Dorrell et al, 2016). Thus, we sought to identify robust, reproducible endocrine cell subpopulations present in our 48-donor cohort and assess if they are significantly altered in T2D, PD, or ND states. We analyzed 74,812 α-, 99,029 β-, and 10,770 δ- cells from T2D, PD, and ND islets and identified eight β-cell subpopulations and seven putative α- and δ-cell subpopulations (Dataset EV6; Appendix Figs. S5a, S6a, and S7a). All β-cell subpopulations expressed *INS* at comparable levels, confirming their β-cell identity. No variation in (sub)clustering between sexes, ancestries, or sequencing chemistries was observed (Appendix Figs. S5b–d, S6b–d, and S7b–d), and each cell type subpopulation was detected in every donor across all glycemic states (Dataset EV6).

Each pancreatic islet endocrine cell type contained putative subpopulations with gene expression signatures of endoplasmic reticulum (ER) stress response signatures-cluster 7 (α) and cluster 6 (β) - and hypoxia-cluster 2 (α), cluster 2 (β), and cluster 4 (δ)

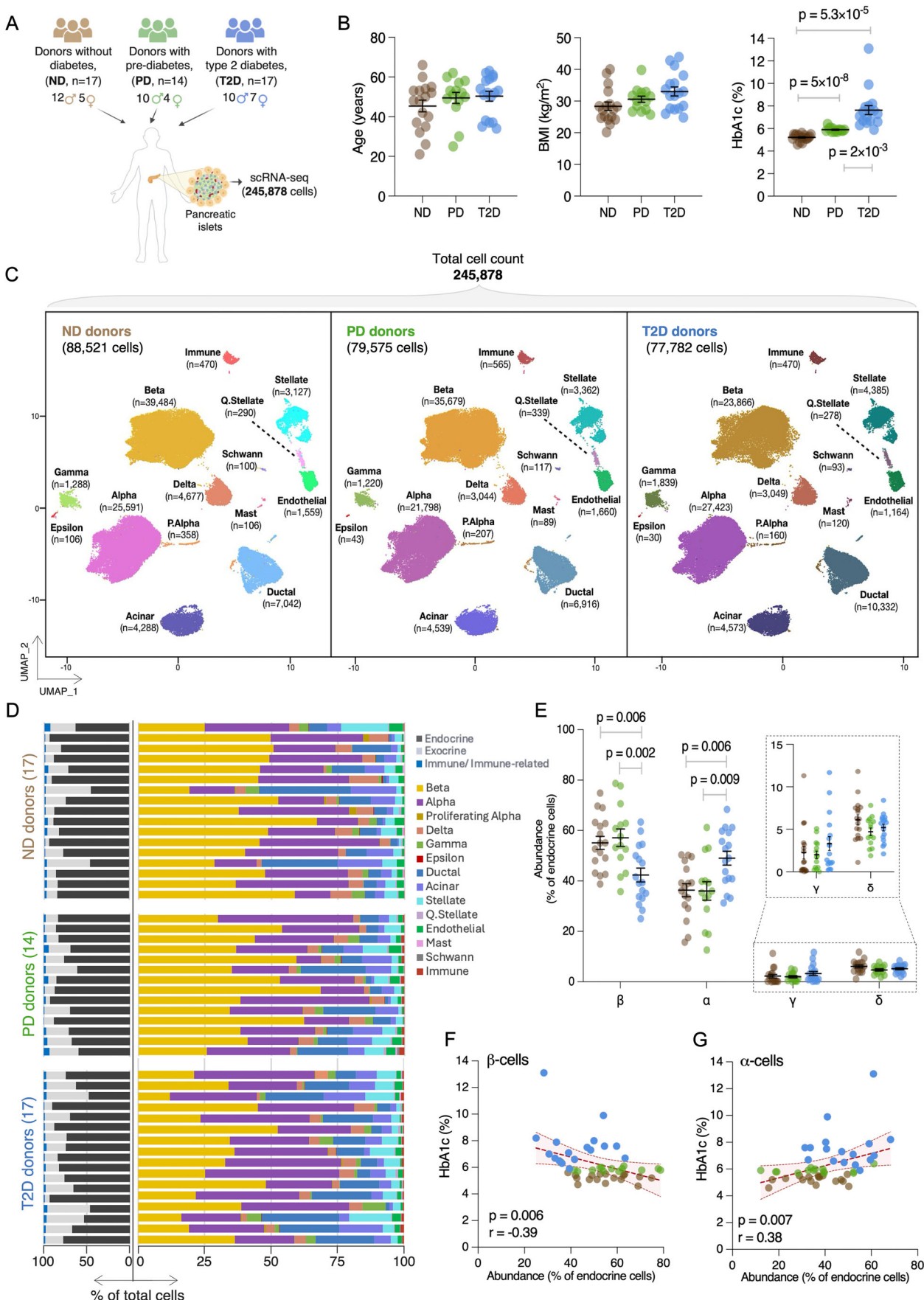

◀ **Figure 1. Human pancreatic islet single-cell transcriptomes from 48-donor cohort reveal cell type proportion variability in T2D donors.**

(A) Human pancreatic islets from 48 cadaveric donors—17 with diagnosed type 2 diabetes (T2D, blue), 14 with HbA1c-based prediabetes (PD; HbA1c 5.7%-6.4%, green), and 17 without diabetes (ND, tan)—were dissociated into single cells and profiled using droplet-based scRNA-seq to obtain 245,878 high-quality islet single-cell transcriptomes. (B) Comparison of age, body mass index (BMI, a measure of obesity), and glycated hemoglobin (HbA1c) between T2D ($n = 17$), PD ($n = 14$), and ND ($n = 17$) individuals in the cohort. Each dot represents an individual donor. The black line and error bars represent the mean ± standard error of the mean. Significant differences ($P < 0.05$, Games–Howell post-hoc test) are reported. (C) Uniform Manifold Approximation and Projection (UMAP) plots displaying unsupervised clustering of 245,878 cells from ND (left), PD (middle), and T2D (right) donors reveal 14 distinct cell types based on the expression of the 2000 most variable genes across the cells. $n$=number of single-cell transcriptomes obtained for each cell type. (D) Relative percentages of endocrine, exocrine, and immune (left) and specific (right) cell types profiled in per-donor stacked bar plots across the glycemic states. Note reduced β-cell proportions in islets from T2D ($n = 17$, bottom) vs. PD ($n = 14$, middle) or ND ($n = 17$, top) donors. (E) Relative abundance of α-, β-, δ-, and γ-cells in ND ($n = 17$), PD ($n = 14$), or T2D ($n = 17$) donors. Dots represent the percentage of endocrine cells detected for each donor. Epsilon (ε) cells were rare (0.09%) in all donors and omitted from comparison. The black line and error bars represent the mean ± standard error of the mean. Significant differences ($P < 0.05$, Tukey's honest significance test) are indicated. (F, G) Spearman correlations between HbA1c levels ($y$ axis) and relative β-cell (F, $x$ axis) or α-cell (G, $x$ axis) abundance for all cohort donors ($n = 48$). Bands enclosing the linear regression line represent 99% confidence intervals. Dots represent individual donors colored as in (A) based on their glycemic status. Source data are available online for this figure.

(Dataset EV6; Appendix Figs. S4e, S5e, and S6e). This implies a broader role of ER homeostasis perturbations or cycles in multiple islet cell types (Maestas et al, 2024) beyond β-cells (Appendix Fig. S8) (Sharma et al, 2015; Baron et al, 2016; Dominguez-Gutierrez et al, 2019). In addition, we detected a putative β-cell subpopulation exhibiting elevated expression of genes enriched in the "insulin secretion" pathway that included genes linked to monogenic or type 2 diabetes, such as *ABCC8, G6PC2, PDX1, SLC30A8, RBP4* (cluster 1). We also identified β-cell clusters with expression signatures associated with translation initiation (cluster 3); heat shock proteins (cluster 4, proliferative vs. mature cells with elevated *CFAP126* (*Fltp* in mice) expression (Bader et al, 2016)); regulation of signaling receptor activity (cluster 5, CD63$^{hi}$ cells with enhanced glucose-stimulated insulin secretion (Rubio-Navarro et al, 2023)); cellular senescence (e.g., *CDKN2A, CDKN2B, PLK2, B2M* expression, cluster 7); and cellular transport (cluster 8) (Dataset EV6).

α- and δ-cell subclusters showed no significant difference between T2D, PD, or ND donors (Appendix Figs. S6f and S7f). All β-cell subpopulations were detected in ND, PD, and T2D donor islets, but two exhibited significant reciprocal quantitative differences in their relative abundance between T2D vs. ND or PD donors (Appendix Fig. S5f). The cluster 1 subpopulation, with elevated expression of genes involved in insulin secretion, was reduced by an average of 10.5% in T2D vs. ND β-cells ($P = 0.001$). In contrast, the proportion of "cellular senescence" cluster 7 cells increased by an average of 12.3% in T2D vs ND β-cells ($P = 0.009$); this significant increase was also observed in T2D vs. PD β-cells ($P = 0.02$, 9.7% average increase). These unsupervised subpopulation analyses thus support emerging reports of increased β-cell senescence in T2D (Aguayo-Mazzucato et al, 2019; Cha et al, 2023; Sone and Kagawa, 2005). Together, these subpopulation shifts, combined with 10–15% overall reductions in T2D donor β-cell numbers/proportions in this cohort (Fig. 1E), implicate ~25–30% reduction of functional β-cells in T2D vs. ND or PD donors.

## Altered β-cell type-specific gene expression and pathway dysregulation in T2D islets

To identify robust cell type-specific genes and pathways that differed between ND, PD, and T2D individuals, we aggregated each individual's scRNA-seq profiles per cell type into "pseudobulk" gene expression profiles and compared them ("Methods").

Surprisingly, we identified only three α-, two δ-, and six γ-cell-specific DEGs between the three glycemic states at FDR < 5% (Datasets EV7, EV8, and EV9). Principal component analysis (PCA) of β-cell transcriptomes suggested modest differences in transcriptional profiles of PD vs. ND donors, but only one gene was differentially expressed at FDR < 5% (PD vs. ND, Dataset EV9). These data suggest that, while PD may represent an early transitional state preceding major remodeling observed in T2D, it is represented by subtle expression changes. Alternatively, the data may indicate that islets from PD donors are not significantly altered and that the PD state instead results from or reflects changes in donor insulin resistance. In contrast, we identified 746 β-cell DEGs at FDR < 5% (T2D vs ND, Dataset EV7), ~10x those detected at FDR < 10% in a recent HPAP cohort-based study (Elgamal et al, 2023), likely resulting from greater sequencing depth and increased number of high-quality and endocrine cells captured in this cohort (Appendix Fig. S2a–c). Alternatively, analytical differences, such as cut-off stringency and pipeline parameters, may contribute to some of these differences. 511 β-cell DEGs exhibited ≥50% fold changes in expression, including 316 upregulated and 195 downregulated genes (Fig. 2A). Importantly, in addition to validating 171 T2D DEGs were previously reported by whole islet or cell type-specific studies (Elgamal et al, 2023; Fang et al, 2019; Weng et al, 2023; Segerstolpe et al, 2016; Bacos et al, 2023; Bosi et al, 2020; Wang et al, 2016; Fadista et al, 2014; Marselli et al, 2020; Solimena et al, 2018; Xin et al, 2016), including *FXYD2, SLC2A2, SCN9A, PAX5, DGKB, IRS1* and *SYT1*, two-thirds of detected β-cell DEGs were previously unreported ($n = 340$; Appendix Fig. S9a–d; Dataset EV10). Sample sizes were insufficient for meaningful analysis or interpretation of sex-specific differences in this cohort.

Several top T2D DEGs have not been previously linked to islet β-cell dysfunction. To test their functional relevance, we assessed if shRNA-mediated knockdown of leading downregulated genes (*MPP1, CD82, GLUL,* and *GOLT1A,* Appendix Fig. S10a) affected glucose-stimulated insulin secretion (GSIS) or cell death in human EndoC-βH3 β-cells (Fig. 2B–D). *INS* shRNA knockdown (99%), used as a positive control, reduced stimulation index and intracellular insulin content by 67.8% and 78.4%, respectively, compared to non-targeting (NT) shRNA control, as expected. Similarly, knockdown of *FXYD2*, a widely detected T2D β-cell DEG (Weng et al, 2023; Xin et al, 2016; Solimena et al, 2018; Segerstolpe et al, 2016; Bosi et al, 2020; Bacos et al, 2023; Fang et al, 2019), reduced stimulation index (Fig. 2B) and intracellular insulin

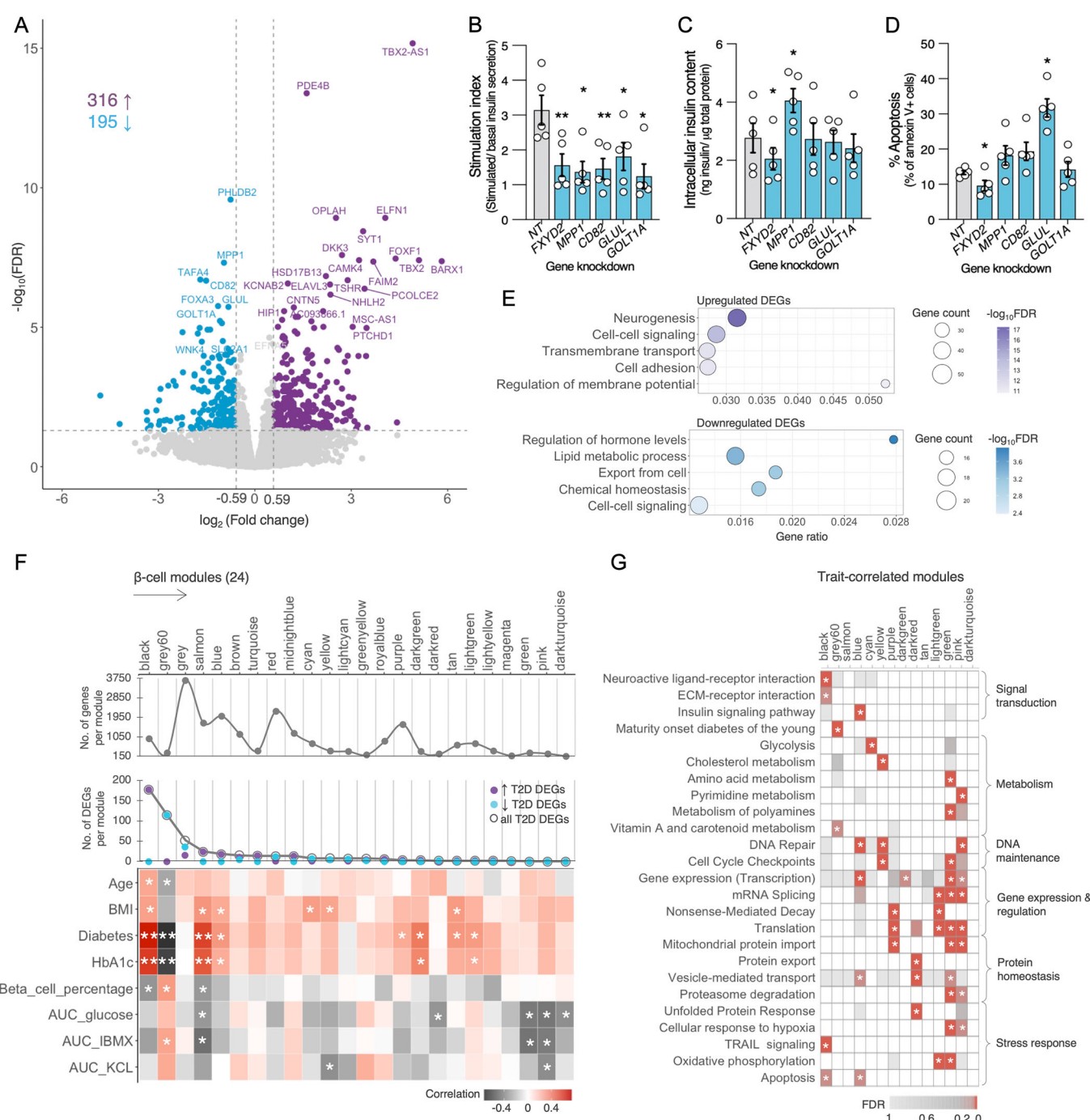

content (Fig. 2C) compared to NT, which is concordant with previous results (Tacto et al, 2025). Knockdown of all candidate genes decreased stimulation index (Fig. 2B), indicating impaired β-cell insulin secretory response to glucose, a T2D hallmark. Additionally, *MPP1* knockdown significantly increased intracellular insulin content (Fig. 2C) and basal insulin secretion (Appendix Fig. S10b). In parallel, we assessed if knockdown of these newly identified T2D DEGs altered β-cell viability using Annexin V/7-AAD staining. In concordance with *Fxyd2* knockout mouse phenotypes showing increases in β-cell mass (Arystarkhova et al,

2013), apoptosis indices were reduced in *FXYD2* knockdown cells compared to the NT shRNA controls (Fig. 2D). Notably, *GLUL* knockdown increased β-cell apoptosis approximately threefold (Fig. 2D; 31.7% vs. 13.3% in NT control cells; $P = 0.004$). *GLUL* encodes glutamine synthetase, which uniquely catalyzes the conversion of glutamate and ammonia to glutamine (Tecson et al, 2025). Matschinsky and colleagues previously demonstrated roles for glutamine in both amino acid- and glucose-stimulated insulin secretion (Li et al, 2004). Although previous studies did not directly link altered *GLUL* expression to T2D, reduced plasma

**Figure 2.  Altered gene expression patterns in T2D β-cells.**

(A) Volcano plot of differentially expressed genes (DEGs) in T2D vs. ND β-cells. Each dot denotes a gene. 511 genes with significant differences in expression at false discovery rate (FDR) < 5% and fold change ≥50% are colored blue (T2D-downregulated) or purple (T2D-upregulated); gray dots denote those with comparable expression in T2D and ND β-cells. (B–D) Effects of shRNA knockdown of newly identified T2D-downregulated β-cell genes on (B) stimulation index (glucose-stimulated/ basal insulin secretion), (C) intracellular insulin content, or (D) apoptosis compared to non-targeting control (NT) in human EndoC-βH3 cells. *FXYD2* was included as a positive control in the assays. Data represent mean ± standard error of the means (s.e.m.) from five biological replicates. *$P < 0.05$, **$P < 0.001$; Student's $t$ test (two-tailed, paired) comparing each target to the NT control (stimulation index: $p_{FXYD2} = 0.0003$, $p_{MPP1} = 0.008$, $p_{CD82} = 0.0001$, $p_{GLUL} = 0.003$, $p_{GOLT1A} = 0.04$; insulin content: $p_{FXYD2} = 0.04$, $p_{MPP1} = 0.008$; apoptosis: $p_{FXYD2} = 0.04$, $p_{GLUL} = 0.004$) (E) Biological process enrichment analysis of upregulated (purple) or downregulated (blue) genes using the molecular signatures database (MSigDB, BROAD Institute). Non-redundant processes with FDR < 5% are displayed. The number of tested upregulated ($n = 316$) or downregulated ($n = 195$) T2D β-cell DEGs relative to the total number of genes in each functional term is provided as a gene ratio. (F) Co-expressed β-cell gene modules in 48-donor cohort using Weighted Gene Co-expression Network Analysis (WGCNA). (Top) Number of genes comprising each module; (Middle) Number of T2D up- (purple) or downregulated (blue) genes from panel a present in each module. (Bottom) Module correlations with donor attributes (Age through HbA1c) or islet composition/functional measures (beta-cell percentage through AUCs). *$P < 0.05$, **$P < 0.001$; color-coded scale indicates Spearman nonparametric correlation for each module-trait pair. BMI body mass index, HbA1c glycated hemoglobin, AUC area under the curve. (G) Pathway enrichment for WGCNA modules. Asterisks indicate pathways enriched at FDR < 5% in gene over-representation testing. White boxes indicate pathways for which no representative genes were in the given WGCNA module and were therefore not tested for enrichment. Source data are available online for this figure.

glutamine and elevated plasma glutamate levels are associated with increased T2D incidence (Liu et al, 2019). Glutamine, a conditionally essential amino acid, supports cell survival by serving as a carbon source for the TCA cycle when glucose-derived pyruvate supplies are compromised (Yang et al, 2014). In T2D islets, reduced *GLUL* expression may weaken this protective metabolic adaptation and contribute to β-cell death by limiting glutamine availability. In addition, excessive glutamate accumulation can induce β-cell toxicity (Huang et al, 2017a) and accelerate apoptosis through sustained activation of glutamatergic *N*-methyl-D-aspartate receptors (NMDARs) (Huang et al, 2017b).

Supporting the importance of neuroactive signaling in T2D β-cell dysfunction, the five most significantly dysregulated protein-coding genes all have neuronal functions, including mediating central nervous system effects of therapeutic agents (*PDE4B*) (Tibbo and Baillie, 2020), acetylcholine receptor aggregation in postsynaptic membranes (*PHLDB2*) (Xie et al, 2019), catalyzing proline conversion to the major excitatory neurotransmitter glutamate (*OPLAH*) (Almaghlouth et al, 2012), regulating post-synaptic neural circuit dynamics (*ELFN1*) (Dunn et al, 2018), and sensing calcium for neurotransmitter release (*SYT1*) (Xu et al, 2009). More broadly, 17.1% of T2D-upregulated β-cell genes were enriched for "neurogenesis" (FDR $q = 6.0 \times 10^{-18}$), followed by "cell–cell signaling" ($q = 1.0 \times 10^{-14}$), "transmembrane transport" ($q = 3.1 \times 10^{-12}$), "cell adhesion" ($q = 5.1 \times 10^{-12}$), and "regulation of membrane potential" ($q = 1.7 \times 10^{-11}$) (Fig. 2E; Dataset EV11). Conversely, downregulated T2D β-cell genes were enriched for "regulation of hormone levels" ($q = 1.2 \times 10^{-4}$), "lipid metabolism" ($q = 4.3 \times 10^{-4}$), and "cell–cell signaling" ($q = 4.2 \times 10^{-3}$, Fig. 2E). MSigDB pathway enrichment analyses revealed "neuroactive ligand–receptor interaction" ($q = 3.9 \times 10^{-4}$) and "vitamin A and carotenoid metabolism" ($q = 1.5 \times 10^{-3}$) as primary molecular pathways associated with the up- and downregulated DEGs, respectively (Dataset EV11).

To complement the differential expression analysis, we used weighted gene co-expression network analysis (WGCNA; "Methods") to identify co-expressed islet gene modules associated with T2D. We identified 24 distinct β-cell gene modules (Fig. 2F; Dataset EV12), each assigned a unique color ID and ranging in size from 3643 genes (gray, largest) to 154 genes (darkturquoise, smallest). Two modules showed significant positive correlations with T2D: the "salmon" and "black" modules (Spearman's $r \geq 0.63$;

$P = 1.54 \times 10^{-6}$), both also correlating with HbA1c (Spearman's $r \geq 0.57$; $P = 2.14 \times 10^{-5}$). The salmon module lacked clear process enrichments (Fig. 2G) and contained few DEGs ($n = 24$), but it included genes encoding proteins enriched in the endomembrane system (Benjamini-Hochberg FDR = $4.3 \times 10^{-4}$; Dataset EV12). The 'black' module contained 56% of upregulated DEGs (178/316) and exhibited modest negative correlations with β-cell percentage (Spearman's $r = -0.36$; $P = 0.03$). This module showed enrichment for signal transduction pathways–"neuroactive ligand–receptor interaction" (FDR = $6 \times 10^{-3}$), "ECM-receptor interaction" (FDR = 0.04), and cellular stress response–"Tumor necrosis factor-related apoptosis-inducing ligand (TRAIL) signaling" (FDR = $8.3 \times 10^{-4}$), "apoptosis" (FDR = 0.03) (Fig. 2G; Dataset EV12), reflecting states of β-cell stress and activation of pro-apoptotic signaling contributing to β-cell loss in T2D.

In contrast, the "gray60" module containing most down-regulated DEGs (114/195) showed significant negative correlations with T2D status (Spearman's $r = -0.74$; $P = 1.7 \times 10^{-9}$) and HbA1c (Spearman's $r = -0.67$; $P = 1.8 \times 10^{-7}$) and modest positive association with β-cell percentage (Spearman's $r = 0.35$; $P = 0.03$) (Fig. 2F; Dataset EV12). This module was enriched for highly penetrant "Maturity onset diabetes of the young (MODY)" genes (FDR = $1.4 \times 10^{-4}$) and altered "Vitamin A metabolism" (FDR = 0.01), crucial for normal β-cell development and function (Fig. 2G and Dataset EV12). These findings reveal two key mechanisms of β-cell dysfunction in T2D—cellular stress response activation and loss of normal β-cell identity—and highlight potential targets for preserving β-cell mass and function in diabetes.

## Dysregulated neuroactive ligand receptor and vitamin A metabolism defects in T2D β-cells

Since complementary WGCNA and DEG analyses both revealed significant enrichment for neuroactive ligand–receptor interaction in T2D-upregulated modules/DEGs and vitamin A metabolism in those that were downregulated, we investigated these pathways further. Upregulated neuroactive ligand–receptor genes included those encoding glutamate (*GRIN1*, *GRIN2A*), acetylcholine (*CHRNA3*, *CHRNA5*), and norepinephrine (*ADRB1*) receptors; neuroendocrine peptides (*GRP*, *GAL*); ATP receptors (*P2RY1*, *P2RX5*); receptors for hormones with known insulin-suppressive effects (*GHSR*, *TSHR*); and a ligand-gated ion channel (*GLRA3*)

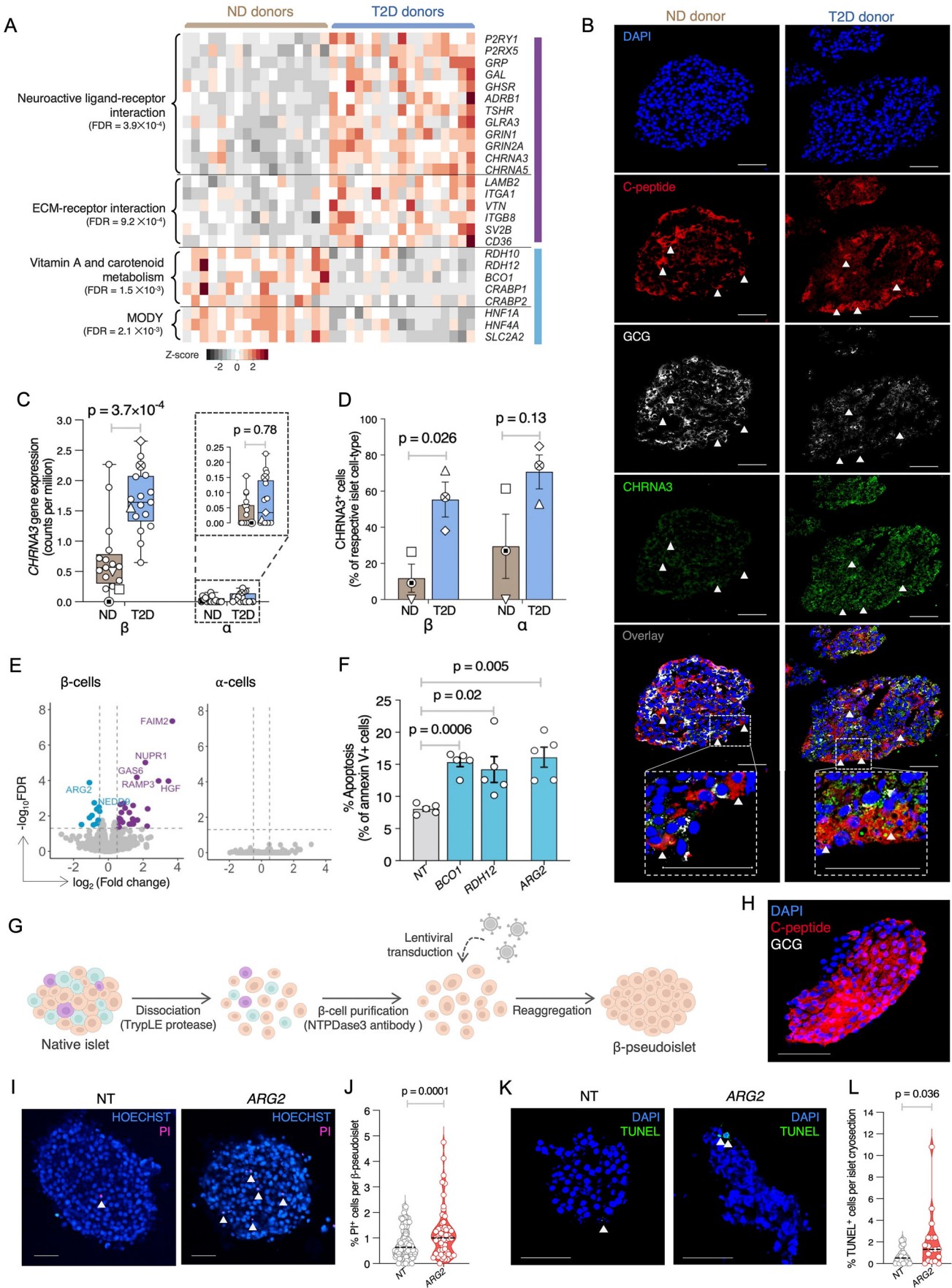

**Figure 3. Validation of up- and downregulated pathway components in T2D β-cells.**

(A) Normalized expression of genes comprising up- (purple rectangle) or downregulated (blue rectangle) pathways in pseudobulk β-cell scRNA-seq profiles from T2D (blue) or ND (tan) donors. (B) Representative immunofluorescence (IF) of CHRNA3 protein levels in ND (left) or T2D (right) donor islet. C-peptide (red) and GCG (white) co-staining was used to identify β- or α-cells in each islet. DAPI (blue) labeled all cell nuclei. Arrowheads highlight CHRNA3⁺(green) β-cells. (C) Boxplots showing normalized (cpm) *CHRNA3* expression in β- or α-cells from pseudobulk ND ($n = 17$) or T2D ($n = 17$) donor islet scRNA-seq profiles. Boxes represent the interquartile range (25th–75th percentiles), the center line indicates the median, and whiskers extend from the minimum to the maximum values. All individual data points are shown. Nominal *P* values from pseudobulk differential gene expression analysis by edgeR (transcriptome-wide analysis) are reported. (D) Percent of CHRNA3-positive islet β- or α-cells based on CellProfiler analysis of IF images (shown in (B)). Percentages were calculated using data obtained from three cryosections, each from three ND vs. three T2D donors (average 756 cells per donor). Shapes denote each islet donor and are provided to compare with their distribution in the paired scRNA-seq data (shown in (C)). Data plotted as mean ± s.e.m. from three ND donors and three T2D donors. *P* values were calculated using Student's *t* test (two-tailed, unequal variance). (E) Volcano plot of differentially expressed cell death genes from T2D vs. ND donor islets in β- (left) or α- (right) cells. FDR-values (*y* axis) are from pseudobulk differential gene expression analysis by edgeR (transcriptome-wide analysis). (F) Apoptosis (assessed via Annexin V staining) in human EndoC-βH3 β-cells after shRNA knockdown of vitamin A/carotenoid metabolism-regulating genes (*BCO1, RDH12*) or a vitamin A target gene (*ARG2*) vs. non-targeting control (NT). Data plotted as mean ± s.e.m. from five biological replicates. *P* values were calculated using Student's *t* test (two-tailed, paired). (G) Schematic showing generation of human β-cell-enriched pseudoislets and shRNA-mediated gene knockdown. (H) Immunofluorescence of C-peptide (red) and glucagon (GCG; white) immunostaining, confirming β-cell enrichment in pseudoislets. DAPI (blue) was used to stain pseudoislet nuclei. (I) Representative Propidium iodine (PI) staining in *ARG2* (right) or non-targeting control (NT, left) shRNA knockdown human β-pseudoislets. (J) Percent cell death (PI⁺/Hoechst⁺) quantification in *ARG2* or NT control shRNA knockdown human β-pseudoislets. Measurements were obtained from an average of 24 islets from each of three islet donors. (K) Representative TUNEL staining images in *ARG2* (right) or non-targeting control (NT, left) shRNA knockdown human β-pseudoislets. (L) Percent apoptosis (TUNEL⁺/DAPI⁺) quantification in *ARG2* or NT control shRNA knockdown human β-pseudoislets. Measurements were obtained from an average of six cryosections from each of three islet donors. *P* values in (J, L) were calculated using a mixed-effects linear regression model with donor specified as a random effect to account for inter-donor variability. Scale bars for all images = 50 microns. Source data are available online for this figure.

(Fig. 3A; Appendix Fig. S11 and Dataset EV11). This upregulation may represent a compensatory response to metabolic stress. Pancreatic islets are densely innervated by autonomic nerves (Alvarsson et al, 2020) releasing acetylcholine (Hampton et al, 2022), norepinephrine, glutamate, and galanin (encoded by *GAL*) (Dunning and Taborsky, 1989), as well as gastrin releasing peptide (encoded by *GRP*) (Agerskov and Nyeng, 2024), which regulate glucagon and insulin secretion. Since neural signals can override circulating glucose effects (Meyers et al, 2016; Stanley et al, 2016), increased neuroreceptor expression in T2D islets might reflect the nervous system's attempt to preserve or augment islet function under metabolic stress (Alvarsson et al, 2020).

Previous studies have reported robust β-cell expression of nicotinic cholinergic receptors, particularly *CHRNA3* (Yoshikawa et al, 2005; Bsharat et al, 2023; Ganic et al, 2016). As proof of principle, we completed CHRNA3 immunostaining to test if protein levels were elevated in T2D vs. ND islets (Fig. 3B). Supporting our RNA-based discovery (Fig. 3C), T2D donor islets exhibited notable increases in CHRNA3-positive β-cells compared to ND (55.4% T2D vs. 11.9% ND; $P = 0.026$) (Fig. 3D; Appendix Fig. S12). CHRNA3 protein was also present in α-cells, consistent with prior reports (Bsharat et al, 2023), but CHRNA3-positive α-cell proportions did not differ significantly between ND and T2D islets (Fig. 3D). In *Ascl1* β-cell-specific knockout mice fed a high-fat diet (Ab et al, 2023), *CHRNA3* induction in β-cells accompanies increased islet innervation and enhances acetylcholine-mediated signaling to promote insulin secretion. Thus, we speculate that upregulation of this and other neuroendocrine signaling genes in T2D islets may represent a compensatory adaptation to sustain insulin secretion in metabolically stressed states. Follow-up studies will determine if it ultimately preserves β-cell function or contributes to maladaptive remodeling and pancreatic dysfunction.

Downregulation of multiple genes encoding key proteins in vitamin A metabolism, including retinol dehydrogenases (*RDH10, RDH12*), β-carotene oxygenase (*BCO1*), and cellular retinoic acid binding proteins (*CRABP1* and *CRABP2*), was another hallmark of

T2D β-cells (Fig. 3A; Appendix Fig. S13a). Vitamin A metabolites, particularly retinoic acid, regulate gene expression by activating transcription factors (e.g., HNF4A, retinoid A receptor nuclear receptor (RAR), retinoid X receptor alpha (RXRA)) that bind to RA-response elements in target genes. Since retinoids regulate apoptosis (Noy, 2010; Lavudi et al, 2023) and vitamin A deprivation decreases β-cell mass (Trasino et al, 2015a), we assessed if compromised vitamin A metabolism contributes to β-cell death. T2D β-cells exhibited differential expression of several genes encoding cell death-associated proteins (Uhlen et al, 2010), including reduction of *ARG2* and *NEDD9* and induction of *FAIM2, NUPR1, GAS6, HGF*, and *RAMP3*, none of which were altered in T2D α-cells (Fig. 3E). Notably, *ARG2* and *NEDD9* promoters harbor response elements for RXRA (Appendix Fig. S14), which is directly activated by the key vitamin A metabolite 9-*cis* retinoic acid (9cRA) (Yoo et al, 2023; Levin et al, 1992).

To test if T2D downregulated vitamin A metabolism pathway genes alter β-cell survival, we completed shRNA-mediated knockdown of *BCO1* (a rate-limiting enzyme in vitamin A metabolism (Coronel et al, 2022)), *RDH12* (involved in 9cRA biosynthesis) (Haeseleer et al, 2002), and *ARG2* (a downstream target of vitamin A metabolism) in human EndoC-βH3 β-cells (Appendix Fig. S13b). Knockdown of all three genes significantly increased β-cell apoptosis relative to NT control (Fig. 3F), demonstrating that downregulation of vitamin A metabolism may contribute to reduced β-cell numbers in T2D. Consistent with these observations, ARG2 protein levels are reduced in human T2D islets (Appendix Fig. S15a) (Ewald et al, 2024), so we tested if *ARG2* shRNA knockdown altered cell death rates in human β-cell-enriched pseudoislets (Fig. 3G,H; "Methods"). Compared to non-targeting shRNA control β-pseudoislets, *ARG2* knockdown (Appendix Fig. S13c) increased cell death, measured by propidium iodide (PI)–Hoechst staining, approximately 1.5-fold (Fig. 3I,J; $P = 0.0001$). TUNEL staining of paired cryo-embedded β-pseudoislets also revealed a significant 2.8-fold increase in apoptosis in *ARG2* shRNA vs. NT control cells (Fig. 3K,L; $P = 0.036$). Together, these results suggest that downregulation of genes in the vitamin A metabolism pathway in T2D reduces β-cell viability in human islets.

## T2D DEG-GWAS integration prioritizes putative causal genes, including *PDZK1*

In addition to directly testing selected DEG effects on human β-cell viability and function, we sought to nominate additional DEGs contributing to, rather than a consequence of, T2D physiology based on their links to glucose homeostasis and T2D genetics. Interestingly, genes harboring inactivating monogenic diabetes mutations, such as *HNF1A*, *HNF4A*, and *SLC2A2* (Zhang et al, 2021), were significantly downregulated in T2D donor β-cells (Fig. 3A). Given these concordant downregulation/loss-of-function effects, we assessed T2D β-cell DEGs for which T2D-associated risk alleles exerted concordant gene expression effects. We compiled a list of 39,972 T2D-associated index and linked proxy genetic variants (all-ancestry or ancestry-specific LD $r^2 \geq 0.80$ in 1000Genomes Phase 3 data) reported in genome-wide association study (GWAS) meta-analyses from T2DGGI (Suzuki et al, 2024), DIAMANTE (Mahajan et al, 2022), MVP (Vujkovic et al, 2020), and AGEN (Spracklen et al, 2020) (Dataset EV13) and queried islet expression QTL association results from the TIGER consortium (Alonso et al, 2021) to identify 461 T2D variants (representing 41 loci) significantly associated with altered islet expression of 25 upregulated or 16 downregulated T2D β-cell DEGs.

Twenty-seven genes exhibited concordant T2D genetic and environmental effects (Fig. 4A, red and blue; Dataset EV14), i.e., the T2D risk allele altered islet gene expression in the same direction as T2D vs. ND β-cell differential expression, including *DGKB*, *ST6GAL1*, and *STARD10*, reported as colocalized T2D genetic and islet eQTL association signals showing directionally consistent impact on gene expression (Alonso et al, 2021; Viñuela et al, 2020; van de Bunt et al, 2015). For nineteen T2D β-cell DEGs, we identified a single variant for which the T2D risk allele altered expression in the same direction. For example, rs67897819 T2D risk allele 'A' (OR = 1.07, $P = 1.7 \times 10^{-68}$) (Suzuki et al, 2024), upstream of *HNF4A*, is associated with decreased expression of *SGK2*, a serum and glucocorticoid inducible kinase gene 800 kb upstream of *HNF4A* (Z-score = −2.02, $P = 0.04$) (Alonso et al, 2021), but not *HNF4A* (Z-score = −0.46, $P = 0.65$) (Alonso et al, 2021). Consistent with the T2D risk allele decreasing *SGK2* expression, it was also a downregulated T2D β-cell DEG (log$_2$FC = −1.36, FDR = $8.5 \times 10^{-4}$). Expression of genes located within 500 kb of rs67897819 (*TOX2*, *OSER1*, *GDAP1L1*, *FITM2*, *TTPAL*, *SERINC3*, *PKIG*, *ADA*, and *KCNK15*) did not differ significantly between T2D and ND β-cells, highlighting *SGK2* as a candidate T2D genetic and pathophysiologic causal/effector gene in this region. SGKs, alongside AKT, are activated downstream of mTORC2 (a regulator of β-cell mass) in response to insulin and phosphorylate bona fide AKT target FOXO1 (Zhou et al, 2021), which serves as an anti-apoptotic signal (Kaiser et al, 2013). *SGK2* encodes a pancreas, liver, and kidney-restricted isoform (Kobayashi et al, 1999) that is linked to PD-L1 signaling (Kong et al, 2023) and inhibits ferroptosis (Cheng et al, 2023). Thus, diminished *SGK2* expression in T2D β-cells may contribute to aberrant activation of multiple cell death pathways.

This approach also nominated *RASGRP1, KCNH6*, and *PDZK1* as candidate causal genes (Fig. 4A, red; Dataset EV14) whose reduced expression contributes to T2D genetic risk and/or

pathophysiology by increasing β-cell susceptibility to pathophysiologic stress and/or enhancing apoptosis propensity. Apoptosis is elevated in *RASGRP1*$^{-/-}$ human embryonic stem cell-derived β-cells (Albanus et al, 2025), and *Kcnh6*$^{-/-}$ mice exhibit increased β-cell ER stress, calcium handling defects, and apoptosis that manifests as impaired glucose tolerance (Lu et al, 2020). *KCNH6* mutations have been identified in hypoinsulinemic/hyperglycemic patients (Hyltén-Cavallius et al, 2017), and the KCNH6-targeting compound berberine has been shown to stimulate insulin secretion (Zhao et al, 2021). *PDZK1* encodes a PDZ-domain containing scaffolding protein that mediates interaction between plasma membrane chloride channels (CFTR and ClC-3B) (Gentzsch et al, 2003) and interacts with Phospholipase C-β3 (PLC-β3) (Kim et al, 2012), a key signaling molecule implicated in the regulation of β-cell insulin secretion (Thore et al, 2005). In other cell types, *PDZK1* has been linked to mitochondrial dysfunction and cellular senescence (Shao et al, 2024) along with apoptosis (Handa et al, 2021; An et al, 2025).

Eight DEGs (Fig. 4A, blue) were linked to independent T2D association signals with both concordant and discordant expression effects. For example, rs28413626 T2D risk allele 'A' (OR = 1.03, $P = 2.1 \times 10^{-18}$) (Suzuki et al, 2024) was associated with higher expression of *PITPNM2* (a membrane-associated phosphatidylinositol transfer protein involved in insulin secretion) (Waselle et al, 2005) in whole islets (Z-score = 2.1, $P = 0.04$) (Alonso et al, 2021), while rs1260294 T2D risk allele 'T' (OR = 1.04, $P = 2.2 \times 10^{-15}$) (Vujkovic et al, 2020), distinct from rs28413626 (LD $r^2 = 0.28$, all ancestries combined) (Myers et al, 2020) was associated with lower islet *PITPNM2* expression (Z-score = −2.18, $P = 0.03$) (Alonso et al, 2021). Such counteracting signals may reside in distinct regulatory elements within a gene, such as promoters, enhancers, or silencers which modulate its expression in diverse ways and lead to opposing effects on gene expression. *PITPNM2* is a downregulated T2D β-cell DEG (log$_2$FC = −0.72, FDR = 0.004) and is one of 21 DEGs for which corresponding protein levels were significantly altered in T2D vs. ND islets (Appendix Fig. S15a; $P = 0.009$ for PITPNM2) (Ewald et al, 2024).

Previous studies for some of the genes identified in these analyses, such as *KCNH6*, support the provocative hypothesis that they are causal contributors to islet dysfunction with druggable therapeutic potential (Lu et al, 2020; Zhao et al, 2021). To test this hypothesis, we evaluated the effect of *SGK2*, *PITPNM2*, and *PDZK1* shRNA knockdown on insulin secretion and cell viability in human EndoC-βH3 β-cells (Fig. 4B–E; Appendix Fig. S15b). *RASGRP1* knockdown, included as a positive control based on its reported role in islet dysfunction (Taneera et al, 2012), significantly increased basal insulin secretion (Fig. 4B; $P = 0.0007$) and elevated intracellular insulin content (Fig. 4D; $P = 0.0005$). *SGK2* knockdown increased basal insulin secretion (Fig. 4B; $P = 0.014$) and intracellular insulin content (Fig. 4D; $P = 0.048$). *PDZK1* knockdown increased basal insulin secretion (Fig. 4B; $P = 0.003$) and reduced stimulation index (Fig. 4C; $P = 0.002$). Knockdown of *SGK2*, *PITPNM2*, and *PDZK1* increased apoptosis (Fig. 4E), with *PDZK1* knockdown showing the most pronounced effect (22.4% vs. 8.1% Annexin V-positive cells in *PDZK1* vs. NT control, $P = 0.0003$).

As mentioned above, *SGK2* was reported as a regulator of β-cell mass (Zhou et al, 2021), so we tested if *PITPNM2* or *PDZK1* shRNA knockdown altered β-cell survival of primary human β-

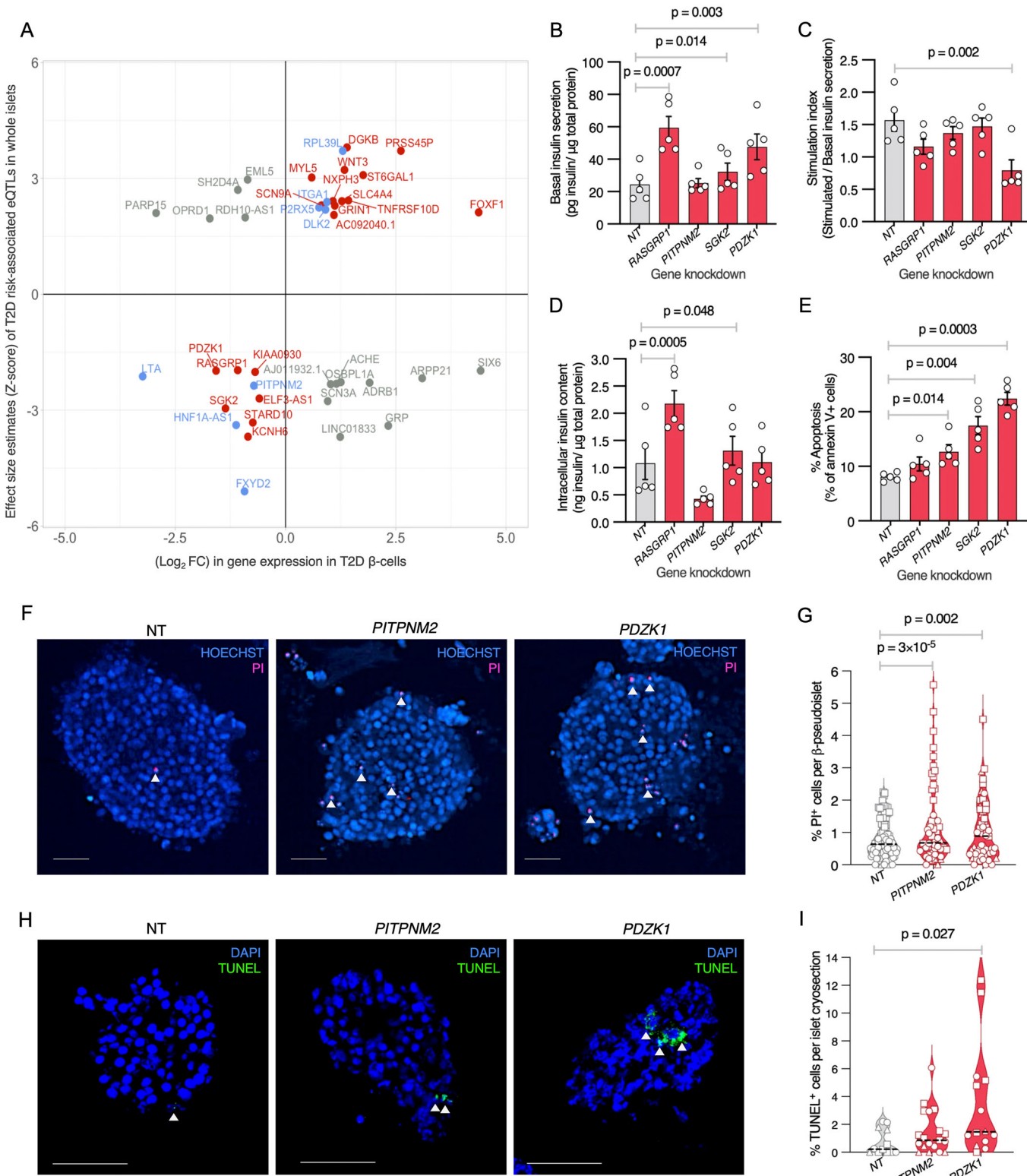

pseudoislets (Fig. 4F–L; Appendix Fig. S15c). *PITPNM2* and *PDZK1* shRNA knockdown significantly increased pseudoislet cell death compared to NT shRNA controls (Fig. 4F,G; $P = 3 \times 10^{-5}$ for *PITPNM2*, $P = 0.002$ for *PDZK1*). Orthogonal assessment of apoptosis in paired cryo-embedded β-pseudoislets (Fig. 4H)

revealed that *PDZK1* knockdown significantly increased the percentage of apoptotic cells compared to NT controls (Fig. 4I; 2.98% vs. 0.72%; $P = 0.027$). Together, these results implicate *PDZK1* as a new regulator of β-cell survival and empirically support its causal role in T2D genetics and pathophysiology.

**Figure 4. Integrated multimodal analyses prioritize T2D β-cell differentially expressed genes as candidate causal/driver genes.**

(A) Comparison of islet eQTL effect sizes from the TIGER consortium (*y* axis) vs. fold change in gene expression for T2D β-cell differentially expressed genes (DEGs, *x* axis) from this study. Red denotes genes with consistent T2D genetic and disease state effects on expression. Upper right quadrant genes are concordantly upregulated; lower left quadrant genes are concordantly downregulated. Gray denotes genes with opposite islet eQTL vs. T2D β-cell differential expression effects. Blue denotes genes with multiple T2D genetic association signals exhibiting both concordant and discordant islet eQTL effects. (B–E) Effects of shRNA knockdown for candidate causal loss-of-function genes on (B) basal insulin secretion, (C) stimulation index, (D) insulin content, or (E) apoptosis in human EndoC-βH3 β-cells compared to a non-targeting control (NT). *P* values were calculated using Student's *t* test (two-tailed, paired) to compare each target gene vs. NT control. (F) Representative Propidium iodine (PI) staining in non-targeting control (NT, left, redisplayed from Fig. 3I), *PITPNM2* (middle), or *PDZK1* (right) shRNA knockdown in human β-pseudoislets. (G) Percent cell death (PI⁺/Hoechst⁺) quantification in *PITPNM2*, *PDZK1*, or NT control shRNA knocked-down human β-pseudoislets. Measurements were obtained from an average of 24 islets from each of three islet donors. (H) Representative TUNEL staining in non-targeting control (NT, left, redisplayed from Fig. 3K), *PITPNM2* (middle), or *PDZK1* (right) shRNA knockdown human β-pseudoislets. (I) Percent apoptosis quantification (TUNEL⁺/DAPI⁺) in *PITPNM2*, *PDZK1*, or NT control shRNA knockdown human β-pseudoislets. Measurements were obtained from an average of 6 cryosections from each of three islet donors. *P* values in (G, I) were calculated using a mixed-effects linear regression model with donor specified as a random effect to account for inter-donor variability. Scale bars for all images = 50 microns. Source data are available online for this figure.

## T2D candidate causal gene GRAMD2B modulates islet viability and glucose homeostasis

To link promising β-cell DEGs to (patho)physiologic effects, we explored diabetes-relevant phenotypes from the International Mouse Phenotyping Consortium (IMPC) (Groza et al, 2023; Meehan et al, 2017), which aims to characterize the function of every protein-coding gene in the mouse genome through whole-body gene knock-out (KO) mouse lines. 198/511 T2D β-cell DEGs were tested by IMPC. Homozygous KO of 35 DEGs in mice caused prenatal lethality ($n = 22$) or sub-lethal fitness phenotypes ($n = 13$), highlighting their roles in essential cell survival processes. Germline deletion of 13 DEGs caused glycemic defects characteristic of T2D etiology (Fig. 5A; Dataset EV15), including *BNIP3* (a mitochondrial protein that can induce apoptosis (Vande Velde et al, 2000) or protect from cell death by removing damaged mitochondria (Burton and Gibson, 2009)) and *GRAMD2B* (a member of ER-plasma membrane contact site family of proteins that regulate intracellular Ca²⁺ dynamics (Besprozvannaya et al, 2018)). Among the 13 DEGs with mouse glycemic phenotypes, GRAMD2B protein levels were lower in T2D islets ($P = 0.002$, Appendix Fig. S16a; Dataset EV14) (Ewald et al, 2024), and *Gramd2b⁻/⁻* archived tissue samples were available at The Jackson Laboratory for analysis (Groza et al, 2023), prompting deeper investigation.

*Gramd2b⁻/⁻* males exhibited impaired glucose tolerance, with AUCs ~1.5-fold higher than WT controls (Fig. 5B; $P = 2.3 \times 10^{-5}$) (Groza et al, 2023), while maintaining normal fasting blood glucose levels (Fig. 5C). Bodyweight trajectories were comparable between *Gramd2b⁻/⁻* and WT mice (Fig. 5D), sustained up to 80 weeks of age (Appendix Fig. S16b). Moreover, fat mass and lean mass were comparable between genotypes (Fig. 5D, insets), indicating that the impaired glucose intolerance in *Gramd2b⁻/⁻* mice is unlikely to be caused by alterations in body composition or obesity-mediated. In contrast, histologic analysis and insulin immunostaining of archived pancreatic tissue sections revealed that *Gramd2b⁻/⁻* mice had smaller islets than age-matched WT controls (Fig. 5E), with a significant 30% reduction in average β-cell area in the mutant mice (Fig. 5F; 0.94% vs.1.35% in WT; $P = 0.003$).

We hypothesized that the reduced islet size and average β-cell area in *Gramd2b⁻/⁻* mice may be due to increased β-cell death. To test this, we assessed β-cell viability following *GRAMD2B* knockdown in human EndoC-βH3 β-cells (Appendix Fig. S16c). Consistent with our hypothesis, *GRAMD2B* shRNA knockdown

significantly enhanced apoptosis relative to NT shRNA control cells (Fig. 5G; $P = 0.039$). Moreover, *GRAMD2B* expression is positively correlated with β-cell proportion in human islets (Fig. 5H, Spearman's $r = 0.42$; $P = 0.003$), supporting a protective role for *GRAMD2B* in β-cell survival. Importantly, reduced *GRAMD2B* expression is orthogonally supported by decreased GRAMD2B protein levels in T2D islets, as demonstrated by proteomic analyses in an independent cohort (Appendix Fig. S16a) (Ewald et al, 2024).

To directly test if *GRAMD2B* inhibition enhances primary human β-cell death, we transduced human β-pseudoislets with *GRAMD2B* shRNA-containing lentivirus (Appendix Fig. S16d) and assessed pseudoislet volume, cell count, and propidium iodide staining using high-throughput imaging and analysis. *GRAMD2B* shRNA knockdown pseudoislets were significantly smaller than NT controls ($P = 5.8 \times 10^{-6}$; $1.0 \times 10^{-6}$ μm³ average decrease in volume (Fig. 5I). These smaller pseudoislets also featured a lower number of Hoechst⁺ nuclei per pseudoislet ($P = 4.0 \times 10^{-5}$, Fig. 5J; Appendix Fig. S16e), indicative of reduced cell density or survival. Consistently, *GRAMD2B*-knocked-down pseudoislets showed a higher percentage of PI⁺ cells ($P = 0.001$), reflecting more dead cells, in agreement with the increased apoptosis in EndoC-βH3 β-cells upon *GRAMD2B* shRNA knockdown (Fig. 5G). Although TUNEL staining revealed a trend toward increased apoptosis in *GRAMD2B* shRNA knockdown pseudoislets, this difference did not reach statistical significance (Appendix Fig. S16f,g).

Together, the multimodal, integrative genetic and phenotypic prioritization framework nominated 58 T2D β-cell DEGs as candidate causal genes. Experimental follow-up of *ARG2*, *PDZK1*, *PITPNM2*, and *GRAMD2B* provide empiric evidence implicating them as modulators of β-cell viability, and we posit the additional prioritized candidates will also exhibit causal effects on islet β-cell viability or function. Future studies of these candidates should provide mechanistic insights into their roles in islet dysfunction and serve as new potential drug targets to prevent or treat β-cell loss in T2D.

## Discussion

We report comprehensive single-cell transcriptomic profiling and comparative analyses of ~250,000 human islet cells from a unique, HPAP-independent 48-donor cohort spanning ND, PD, and T2D states. Consistent with previous studies (Wu et al, 2021; Wang et al, 2023), we detected significant T2D-associated changes in β-cell (sub)

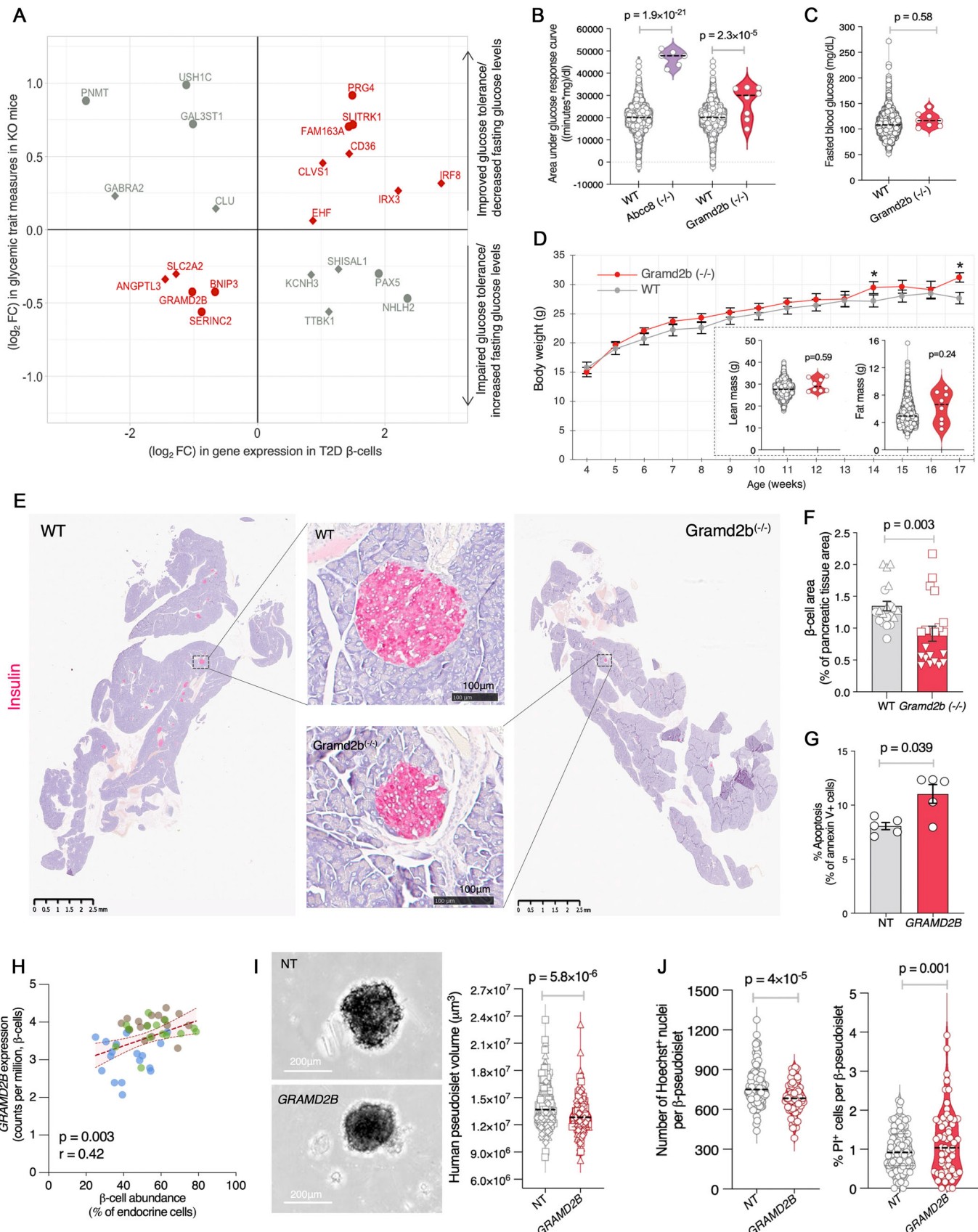

◀ **Figure 5.** *GRAMD2B is a causal T2D candidate gene that promotes glucose homeostasis and β-cell viability.*

(A) T2D β-cell DEGs are significantly associated with various glycemic phenotypes in whole-body knockout (KO) mice from the International Mouse Phenotyping Consortium (IMPC) (Groza et al, 2023). Log$_2$fold change (FC) gene expression in T2D vs. ND β-cells (x axis) compared to Log$_2$fold change in glycemic trait measurements in KO mice vs. wild-type mice (y axis). Circles or diamonds distinguish glucose tolerance (area under glucose response curve and initial response to glucose challenge) vs. fasting glucose phenotypes, respectively. Red denotes genes for which KO glycemic phenotypes consistent with T2D physiology; gray denotes those with apparent T2D-discordant effects. (B) IMPC glucose homeostasis phenotypes of male *Gramd2b*$^{-/-}$ mice (n = 8) vs. wild-type (n = 2704) controls; IMPC phenotype of monogenic diabetes gene *Abcc8*$^{-/-}$ mice (n = 6) is shown for comparison. Glucose tolerance measured by area under the glucose response curve from an intraperitoneal glucose tolerance test (IPGTT). P values were calculated using a linear mixed model and reported as provided by the IMPC. (C) Fasted blood glucose in male *Gramd2b*$^{-/-}$ (n = 8) vs. WT (n = 2706) controls. P values were calculated using a linear mixed model and reported as provided by the IMPC. (D) Body weights of *Gramd2b*$^{-/-}$ (red, n = 6–8) or WT control (gray, n = 1404–2825) males. *P < 0.05 (p$_{14weeks}$ = 0.026, p$_{17weeks}$ = 0.003, two-way ANOVA). Insets show lean and fat mass in 17-week-old males for each genotype. P values were calculated using a linear mixed model and reported as provided by the IMPC. (E) Insulin immunostaining in WT (left) or *Gramd2b*$^{-/-}$ (right) pancreas. Insets show magnified images of a representative islet from each genotype, revealing decreased β-cell area in KO mice. (F) β-cell area % (calculated relative to total pancreatic tissue area in WT or KO sections) assessed via automated analysis of 10 sections from each of two mice per genotype (n = 2, denoted by shapes) is shown as mean ± s.e.m. P value was calculated using Student's t test (two-tailed, unequal variance). (G) Apoptosis (percent Annexin V-positive cells) in *GRAMD2B* vs. non-targeting control (NT) shRNA knockdown human EndoC-βH3 β-cells. Data represent mean ± s.e.m. from five biological replicates. P values were calculated using Student's t test (two-tailed, paired). (H) Correlation between β-cell *GRAMD2B* expression (y axis) and β-cell abundance in 48-donor islet cohort. r = Spearman correlation coefficient. (I) Representative brightfield images of NT or *GRAMD2B* shRNA transduced β-cell-enriched human pseudoislets; violin plot to the right summarizes the 3D volume of shRNA transduced pseudoislets (μm³) from 3 ND donors. Measurements were made for an average of 49 islets from each of three islet donors. The black dashed line indicates median volumes. (J) Violin plots of nuclei counts via Hoechst staining (left) or percent PI$^+$ cells (right) in NT or *GRAMD2B* shRNA-transduced β-pseudoislets from 3 ND donors. Measurements were made for an average of 27 islets from each of three islet donors. The black dashed line indicates the median. Two data points are higher than the plotted y axis limit in the PI$^+$ plot. P values in (I, J) were calculated using a mixed-effects linear regression model with donor ID as a random effect to account for inter-donor variability. Source data are available online for this figure.

populations, collectively manifesting as a ~25–30% reduction in functional β-cell proportions compared to ND or PD donors. We identified 511 genes with perturbed expression in T2D β-cells, 58 of which we nominate as high-priority candidate causal genes based on their convergent and concordant regulation by both genetic and pathophysiologic T2D factors. Follow-up experiments and analyses demonstrate that *PITPNM2, PDZK1,* and *GRAMD2B* perturbation impairs β-cell viability and islet (dys)function, establishing their primary links to T2D pathophysiology and providing proof-of-concept data to support and motivate future studies of the additional causal candidate genes identified. To facilitate further research, we provide these human islet single-cell transcriptomic data in two accessible formats: (1) CellxGene (https://cellxgene.cziscience.com/collections/58e85c2f-d52e-4c19-8393-b854b84d516e) and (2) Transcriptome Atlas of Pancreatic Islet Cells (TAPIC; https://thejacksonlaboratory.shinyapps.io/TAPIC_Stitzel_Lab/).

Unsupervised single-cell transcriptomic analyses across islet cell types revealed eight putative β-cell subpopulations, consistently detected in all donors of this cohort. Several subpopulations correspond to the ones previously reported (Talchai et al, 2012; Sharma et al, 2015; Baron et al, 2016; Dominguez-Gutierrez et al, 2019; Dorrell et al, 2016; Grün et al, 2016; Bader et al, 2016; Rubio-Navarro et al, 2023), including those with elevated expression of *CD63* (Rubio-Navarro et al, 2023) and *CFAP126* (the human *Fltp* orthologue) (Bader et al, 2016). We confirm and extend the presence of the widely reported endoplasmic reticulum (ER) stress signature—previously attributed to β-cell sub-populations (Muraro et al, 2016; Fonseca et al, 2011; Xin et al, 2018)—to α- and δ-cells, suggesting this putative subpopulation may reflect a broader, cyclical state of hormone synthesis and secretion in each islet cell type. However, the relative abundance of these subpopulations did not differ between T2D, ND, or PD individuals, implying they may contribute to physiologic rather than pathologic β-cell heterogeneity. In contrast, we observed a significant T2D-associated increase in β-cells exhibiting a senescence-associated gene expression signature. Senescence has recently been implicated as a (mal)adaptive β-cell process in both type 1 (Lee et al, 2023) and type 2

diabetes (Carapeto et al, 2024), and this subpopulation may underlie the intra-individual heterogeneity in *CDKN2A* expression and chromatin accessibility reported in trajectory-based analyses of T2D β-cells from a smaller cohort (Weng et al, 2023). Further mechanistic studies are warranted to investigate the role of this putative senescent β-cell subpopulation, including the effects of senolytic versus senomorphic agents on islet function in T2D. More broadly, spatial, in situ-based approaches will be critical to assess the organization and functional impact of these subpopulations—both within and between islets—across distinct regions of the pancreas (e.g., head, neck, body, tail) in healthy versus diabetic individuals.

Using pseudobulk analyses, we identified 746 T2D β-cell DEGs at FDR < 5%, with 511 showing >1.5-fold expression changes. Only one-third of these DEGs have been previously reported, including genes such as *DGKB, ASCL2, FXYD2, ARG2,* and *PPP1R1A*. Genes do not function in isolation but operate within coordinated gene networks. Weighted Gene Co-expression Network Analysis (WGCNA) enabled the identification of two distinct yet complementary gene modules, both converging on β-cell failure as a core feature of T2D pathology: the "black" module, reflecting β-cell stress, and the "gray60" module capturing loss of β-cell identity. Together, these findings reinforce the growing consensus around a reproducible set of high-confidence T2D-associated DEGs and perturbed pathways. Continued efforts with larger cohorts and high-quality, standardized processing—such as those led by the Pancreas Knowledgebase (PanKBase, https://pankbase.org/)—should further enhance our multi-omic understanding of islet cell type-specific (dys)function.

Both differential expression and co-expression module analyses identified Vitamin A metabolism as a significantly enriched downregulated pathway/process in T2D β-cells. Complementary studies in model systems have reported links between vitamin A metabolism and islet development and function. For example, Cnop and Pipeleers noted the protective effects of vitamin A vs. LDL-induced toxicity in rat islet β-cells (Cnop et al, 2002). Chemical screens and mechanistic studies in zebrafish revealed an essential requirement for retinoic acid signaling in islet

differentiation and β-cell regeneration (Rovira et al, 2011; Huang et al, 2014, 2016). In rodents, vitamin A deficiency increased β-cell apoptosis, increased α-cell mass, and hyperglucagonemia, and dominant-negative RAR-α mutants caused age-dependent decreases in plasma insulin resulting from impaired GSIS, decreased β-cell mass, and reduced β-cell insulin content (Brun et al, 2015). Dietary vitamin A deficiency has been linked with hyperglycemia, mirroring reduced vitamin A levels in the pancreas (Trasino et al, 2015a). Pancreas β-cells generate 9cRA (an active vitamin A metabolite and a high-affinity ligand for RXR) (Kane, 2012), and β-cell mass reduction (e.g., heterozygous Akita and streptozotocin-treated mice models) is accompanied by proportional decreases in 9cRA levels (Kane et al, 2010). In β-cells, elevated glucose concentrations suppress 9cRA biosynthesis (Yoo et al, 2023), a central process in vitamin A metabolism, governed by the reductases RDH10 and RDH12. Experimental results we report in human EndoC-βH3 cells, as well as primary islet-derived pseudoislets, underscore the importance of Vitamin A metabolism to human β-cell survival and islet pathology in T2D.

The neuroactive ligand–receptor pathway emerged as significantly enriched among upregulated T2D β-cell DEGs. β-cell function is tightly controlled by the autonomic nervous system (Papazoglou et al, 2022; Makhmutova et al, 2021): neurotransmitters are crucial for regulating insulin secretion (Pan et al, 2022) and are emerging as potential therapeutic targets for diabetes management. β-cell-specific deletion of *Grin1*, encoding NMDAR subunit GluN1, exhibited context-dependent effects on islet GSIS and glucose tolerance in mice (Lockridge et al, 2021), and NMDAR antagonist dextromorphan increased serum insulin levels and improved glucose tolerance in patients with T2D (Marquard et al, 2015). Our data also highlight upregulated ATP receptor genes (*P2RY1*, *P2RX5*) and the underappreciated role of purinergic signaling in islet dysfunction and T2D. Notably, insulin secretory vesicles contain ATP and ADP molecules, which are co-released with insulin during glucose-stimulated exocytosis (Obermüller et al, 2005). These extracellular signaling mediators activate purinergic P2 receptors on the β-cell membrane, whose autocrine effects amplify glucose-induced calcium [Ca$^{2+}$] responses in β-cells (Khan et al, 2014). While we demonstrate that increased *CHRNA3* expression in T2D β-cells is reflected at the protein level in human T2D β-cells via immunostaining, further experimental studies in cellular and mouse models are necessary to link a functional consequence to the increased presence of this receptor in β-cells.

Using a combination of genetic and comparative analyses along with experimental approaches, we nominated 58 T2D β-cell DEGs as prioritized candidate causal T2D genes (26 reported and 32 newly identified, Dataset EV14). This prioritized list includes genes with well-established gain- or loss-of-function T2D GWAS variant effects on islet gene expression, such as *DGKB*, *FXYD2*, *PPP1R1A*, and *SLC2A2*, as well as new candidates that span the variant-to-function gamut, like *PITPNM2*, *SGK2*, or *PDZK1*, and capture concordant multi-study, multimodal effects, such as *GRAMD2B*. Several T2D β-cell DEGs exhibit diabetes-relevant glucose homeostasis phenotypes in knockout mice that are directionally consistent with their dysregulation in T2D vs. ND islets, such that germline knockout of up- or downregulated genes improve (e.g., *SLITRK1*, *IRF8*) or impair (e.g., *GRAMD2B*, *BNIP3*) glucose homeostasis, respectively. This study begins to translate human single-cell associations into experimentally validated, disease-relevant gene targets, which we hope will facilitate and inform future mechanistic work. Based on our studies of *ARG2*, *PITPNM2*,

*PDZK1*, and *GRAMD2B*, we expect that systematic and targeted mechanistic studies of the additional genes prioritized through these integrated, multimodal analyses in primary human islets and using β-cell-specific knockouts will establish their causal roles in islet dysfunction and T2D.

Although we identified robust and extensive gene expression changes in T2D β-cells, we detected surprisingly minimal-to-negligible alterations in T2D islet α− or δ-cells. Although the biological vs. potential technical basis for the surprising lack of differences in the other cell types is unclear, it is noteworthy that this apparent conundrum was also observed in recent independent, HPAP cohort-based analyses (Elgamal et al, 2023). All donor islets in this study were handled and processed using standardized protocols after overnight recovery from shipping with minimal additional ex vivo culture, and islet cell types were captured in parallel with comparable sequencing depth, so it is unlikely this discrepancy results from capture or sequencing bias in the platform. Despite best efforts to match donor characteristics, these experiments and analyses involve ex vivo islets from donors with inherently variable lifestyles and environments. It is possible that 5.5 mM glucose concentrations in standard culture conditions may provide stimulation or stress that "unmasks" the β-cell differences and deficiencies observed, but not those in other cell types. For example, α-cells respond to low (0 mM) glucose and amino acids (Armour et al, 2023; Quesada et al, 2008), so islets may need to be cultured in these conditions to "unmask" α-cell deficiencies in T2D islets. Future studies comparing T2D vs. ND islet cell differences with cell type-specific stimuli/stressors will be important to test this hypothesis and could enhance our understanding of non-β-cell contributions to islet dysfunction in T2D. Nonetheless, this study provides new genomic resources, prioritized candidate causal genes with proof-of-concept validation, and mechanistic insights for human islet dysfunction in T2D pathophysiology.

# Methods

**Reagents and tools table**

| Reagent/resource | Reference or source | Identifier or catalog number |
|---|---|---|
| **Experimental models** | | |
| Cadaveric human pancreatic islets | IIDP, Prodo Labs | http://iidp.coh.org/ |
| HEK 293 T | ATCC | Cat# CRL-3216 |
| EndoCβH3 | Univercell Biosolutions | contact@humancell design.com |
| **Recombinant DNA** | | |
| psPAX2 | Addgene | Cat# 12259 |
| pCMV-VSV-G | Addgene | Cat# 8454 |
| **Antibodies** | | |
| Monoclonal mouse anti-human NTPDase3 antibody | info@ectonucleotid ases-ab.com | Cat# hN3-B3S; hN3-H10S |
| C-peptide | Developmental Studies Hybridoma Bank (DSHB) | Cat# GN-ID4 |
| Glucagon | Cell Signaling | Cat# 2760 |
| CHRNA3 | Thermo Fisher | Cat# MA5-31685 |

| Reagent/resource | Reference or source | Identifier or catalog number |
|---|---|---|
| Goat anti-Rat IgG Secondary Antibody | Thermo Fisher | A-11077 |
| Goat anti-Rabbit IgG Secondary Antibody | Thermo Fisher | A-11009 |
| Goat anti-Mouse IgG Secondary Antibody | Thermo Fisher | A-21236 |
| **Oligonucleotides and other sequence-based reagents** | | |
| TaqMan *INS* | Thermo Fisher | Hs00355773_m1 |
| TaqMan *ACTB* | Thermo Fisher | Hs01060665_g1 |
| TaqMan *GOLT1A* | Thermo Fisher | Hs00893792_m1 |
| TaqMan *MPP1* | Thermo Fisher | Hs00609971_m1 |
| TaqMan *GLUL* | Thermo Fisher | Hs00365928_g1 |
| TaqMan *CD82* | Thermo Fisher | Hs01017982_m1 |
| TaqMan *FXYD2* | Thermo Fisher | Hs00253715_m1 |
| Taqman *ARG2* | Thermo Fisher | Hs00982833_m1 |
| Taqman *BCO1* | Thermo Fisher | Hs01015945_m1 |
| Taqman *RDH12* | Thermo Fisher | Hs00288401_m1 |
| Taqman *PDZK1* | Thermo Fisher | Hs00275727_m1 |
| Taqman *PITPNM2* | Thermo Fisher | Hs01096388_m1 |
| Taqman *RASGRP1* | Thermo Fisher | Hs00996734_m1 |
| Taqman *SGK2* | Thermo Fisher | Hs00367639_m1 |
| Taqman *GRAMD2B* | Thermo Fisher | Hs04935481_m1 |
| **Chemicals, enzymes, and other reagents** | | |
| 10x scRNAseq | https://assets.ctfasshow [QJ]sets.net/an68im79xiti/1eX2FPdpeCgnCJtw4fj9Hx/7cb84edaa9eca04b607f9show [QJ]193162994de/CG000204_ChromiumNextGEMSingshow [QJ]leCell3_v3.1_Rev_D.pdf | |
| StemPro™ Accutase™ Cell Dissociation Reagent | Life Technologies | Cat# A1110501 |
| Prodo Media | Prodo labs | Cat# PIM-S001GMP Cat# PIM-ABS001GMP Cat# PIM-G001GMP |
| Advanced DMEM/F-12 | Gibco | Cat# 12634028 |
| 2-beta mercaptoethanol | Gibco | Cat# 21985023 |
| nicotinamide | Sigma-Aldrich | Cat# 72340 |
| Sodium selenite | Sigma-Aldrich | Cat# S1382 |
| Puromycin | Takara | Cat# 631306 |
| Penicillin/Streptomycin | Gibco | Cat# 15140163 |
| BSA | Sigma-Aldrich | Cat# A7030 |

| Reagent/resource | Reference or source | Identifier or catalog number |
|---|---|---|
| Fibronectin | Sigma-Aldrich | Cat# F1141 |
| ECM | Sigma-Aldrich | Cat# E1270 |
| CMRL1066 Media | Gibco | Cat# 11530037 |
| Glutamax | Gibco | Cat# 35050061 |
| HEPES | Gibco | Cat# 15630080 |
| SODIUM PYRUVATE | Gibco | Cat# 11360070 |
| iCell Endothelial Cells Medium Supplement | Fujifilm Cellular Dynamics | Cat# M1019 |
| VascuLife VEGF Medium Complete Kit | LifeLine Cell Technology | Cat# LL-0003 |
| Human insulin ELISA | Mercodia | Cat# 10-1113-10 |
| BCA protein assay kit | PIERCE | Cat# PI-23225 |
| FITC Annexin V Apoptosis Detection Kit with 7-AAD | Biolegend | Cat# 640922 |
| *INS* shRNA | SIGMA | TRCN0000084048 |
| *FXYD2* shRNA | SIGMA | TRCN0000042951 |
| *GOLT1A* shRNA | SIGMA | TRCN0000127937 |
| *MPP1* shRNA | SIGMA | TRCN0000315332 |
| *GLUL* shRNA | SIGMA | TRCN0000343991 |
| *CD82* shRNA | SIGMA | TRCN0000038033 |
| *ARG2* shRNA | SIGMA | TRCN0000051019 |
| *BCO1* shRNA | SIGMA | TRCN0000064723 |
| *RDH12* shRNA | SIGMA | TRCN0000028574 |
| *PDZK1* shRNA | SIGMA | TRCN0000059668 |
| *PITPNM2* shRNA | SIGMA | TRCN0000029764 |
| *RASGRP1* shRNA | SIGMA | TRCN0000048268 |
| *SGK2* shRNA | SIGMA | TRCN0000002112 |
| *GRAMD2B* shRNA | SIGMA | TRCN0000137483 |
| Non-Target shRNA Control | SIGMA | SHC016 |
| RNA to CT kit | Applied Biosystems | Cat# 4392938 |
| LentiX concentrator | Takara | Cat# 631232 |
| LentiX Takara p24 ELISA | Takara | Cat# 632200 |
| Quibit RNA HS Assay | Applied Biosystems | Cat# Q32852 |
| ReadyProbes™ Cell Viability Imaging Kit, Blue(Hoechst 33342)/Red(PI) | Thermo fisher | Cat# R37610 |
| ApopTag PLUS In Situ Apoptosis Detection Kit | SIGMA | Cat# S7111 |
| ProLong™ Diamond Antifade Mountant with DAPI | Thermos Fisher | Cat# P36962 |
| **Software** | | |
| FlowJo | https://flowjo.com BD Sciences | |

| Reagent/resource | Reference or source | Identifier or catalog number |
|---|---|---|
| CellProfiler v4.2.8 | https://cellprofiler.org/ | |
| Harmony | https://www.revvity.com/product/harmony-5-2-office-revvity-hh17000019 | |
| Cellranger v6.1.2 | https://www.10xgenomics.com/support/software/cell-ranger/latest | |
| STARsolo with using STAR 2.7.9a | https://github.com/alexdobin/STAR/blob/master/docs/STARsolo.md Kaminow et al, 2021 | |
| Seurat v4.0 | https://satijalab.org/seurat/articles/get_started.html Hao et al, 2021 | |
| Demuxlet | https://github.com/statgen/demuxlet Kang et al, 2018 | |
| verifyBamID v2 | https://github.com/Griffan/VerifyBamID Zhang et al, 2020 | |
| plink v1.90 | https://www.cog-genomics.org/plink/ Chang et al, 2015 | |
| HRC-1000G-check-bim | https://www.chg.ox.ac.uk/~wrayner/tools/ | |
| bcftools v1.11 | https://samtools.github.io/bcftools/bcftools.html | |
| soupX v1.6.2 | https://github.com/constantAmateur/SoupX Young and Behjati, 2020 | |
| Scrublet v0.2.3 | https://github.com/Moonerss/scrubletR Wolock et al, 2019 | |
| Harmony v3.8 | https://github.com/immunogenomics/harmony Korsunsky et al, 2019 | |
| edgeR v3.34.1 | https://bioconductor.posit.co/packages/3.19/bioc/html/edgeR.html Robinson et al, 2010 | |
| limma v3.66.0 | https://bioconductor.org/packages/release/bioc/html/limma.html Ritchie et al, 2015 | |
| WGCNA v1.74 | https://cran.r-project.org/web/packages/WGCNA/index.html Langfelder and Horvath, 2008 | |
| GSEA v4.4.0 and associated MSigDB | https://www.gsea-msigdb.org/gsea/index.jsp Subramanian et al, 2005 | |
| variancePartition v1.40.1 | https://www.bioconductor.org/packages/release/bioc/html/variancePartition.html Hoffman and Schadt, 2016 | |
| **Other** | | |
| 10x Chromium | https://www.10xgenomics.com | |

## Islet procurement and processing

Pancreatic islets from 48 individuals, consisting of 17 ND, 17 T2D, and 14 PD donors, were received through IIDP or Prodo labs (Table EV1). Because the islets are from de-identified cadaveric organ donors, the study was determined to be human subjects exempt. Only donor islet samples with reported purity ≥80% (median = 90%; range = 80–98%) and viability >90% were accepted for inclusion in this study. Upon receipt from IIDP or ProdoLabs, islets were recovered overnight in ProdoLab media following the ProdoLabs protocol (https://prodolabs.com/protocols/). Islets were dissociated into a single-cell suspension by incubating in Accutase at a ratio of 1000 IEQ per ml for 8 min at 37 °C with trituration every 2 min, as reported before (Lawlor et al, 2017). Depending upon the reported islet index for each IIDP/ProdoLab isolation, we used 200–1000 IEQs per donor for dissociation into a single-cell suspension. Islet cell viability after dissociation was measured using a Countess automated cell counter (Invitrogen), with median post-dissociation viability = 77% (range = 59–89%; Table EV1).

## Single-cell RNA sequencing

Dissociated cells were washed twice and suspended in PBS containing 0.04% BSA, then filtered through a 40-μm Flowmi cell strainer to remove clumped cells and immediately processed as follows. Cell viability was assessed on a Countess automated cell counter (Invitrogen), and 12,000 cells from each suspension were loaded onto one lane of a 10x Genomics Chip G. Single-cell capture, barcoding, and library preparation were performed using the 10X Chromium platform (https://www.10xgenomics.com) according to the manufacturer's protocol for chemistries v2 (#CG00052) and v3 (#CG000183). cDNA and libraries were checked for quality using Tapestation 4200 (Agilent) and Qubit Fluorometer (ThermoFisher), quantified by KAPA qPCR, and sequenced on an Illumina NovaSeq 6000 (S1, S2 or S4 100 cycle flow cell), targeting 6000 barcoded cells with an average sequencing depth of 50,000 reads per cell. Illumina base call files for all libraries were converted to FASTQs using CellRanger-6.1.2 demultiplexing and count pipelines (https://www.10xgenomics.com). Initially, we used cellranger's *mkfastq* to demultiplex the raw base call (BCL) files generated by Illumina sequencers, perform adapter trimming, and retrieve the 10-bp length UMI bases to be included in the generated FASTQ files for downstream processing. We processed the raw FASTQs with STARsolo (Kaminow et al, 2021) using STAR 2.7.9a (https://github.com/alexdobin/STAR/blob/master/docs/STARsolo.md). The barcode demultiplexing was done with the default V2/V3 whitelists coming from the CellRanger v.6 installation (https://kb.10xgenomics.com/s/article/115004506263-What-is-a-barcode-inclusion-list-formerly-barcode-whitelist). For each of the Gel bead-in Emulsions (GEMs), we aligned the reads to the full Ensembl human genome GRCh38 (https://ensembl.org/Homo_sapiens/Info/Index) and used the standard STAR spliced read alignment algorithm to assign them into the exonic, intronic, and intergenic groups. We performed error-correction and deduplication of the Unique Molecular Identifiers (UMIs) and quantified the per-cell gene expression to generate the raw UMI data for each library (Dataset EV1). To filter out the empty droplets, we employed the EmptyDrops_CR background model (Lun et al,

2019) that generated the filtered UMI data of $G = 36,601$ genes (both protein-coding and non-coding) and $N^* = 414,082$ cells across the $J = 54$ libraries for downstream processing. The median number of cells across libraries was 7748 with a 25%–75% interquantile range (IQR) of 5891–9133. Reads per cell per donor are included in Dataset EV1. Our data exhibited the typical high-quality features suggested by the 10x's guidelines (CG000329-Rev A, Technical Note): fraction of reads with valid barcodes (ideal: >0.75; our median: 0.98), fraction of UMI bases with Q-score ≥30 (ideal: >0.65; our median: 0.96) and fraction of unique reads in cells (ideal: >0.70; our median: 0.81).

## Experimental design and metadata information

The similarity of the three glycemic states was assessed in terms of the Euclidean dot product $S(a, b) = \{\sum_{i=1}^{z} a_i b_i\} / \{\sqrt{\sum_{i=1}^{z} a_i^2} \times \sqrt{\sum_{i=1}^{z} b_i^2}\}$, where $a$, $b$ are the z-dimensional attribute vectors of a glycemic state pair under comparison (z = 4) and $a_i$, $b_i$ are the ith components of these vectors (one of sex, ancestry, age, and BMI). We generated $J = 54$ 10X libraries containing the data of either single or multiplexed islets. The first 12 islets (in processing date), coming from 4 ND, 1 PD, and 7 T2D donors, were generated with the V2 chemistry. Cells from 12 islets of V3 chemistry had their RNA sequenced across multiple islet-specific or genetically multiplexed islet libraries (Macosko et al, 2015).

## Ambient RNA decontamination by SoupX

Ambient RNA is the pool of mRNA molecules released in the cell suspension likely from stressed or apoptotic cells. It is incorporated into the droplets, resulting in cross-contamination of transcripts between different cell populations. We estimated and removed contamination in individual cells by the SoupX model (Young and Behjati, 2020) using the *soupX-1.6.2*R package from CRAN. We processed each library $j = 1, \dots, 54$ separately. First, we converted STARsolo's raw and filtered UMI data into respective Seurat v.4 objects (Hao et al, 2021) (STAR-Methods) that were subsequently merged into a single SoupX object. Seurat's filtered data were normalized, scaled and clustered with Leuven on the UMAP reduced representation (see "Seurat clustering by library"). The clustering was fed into the SoupX object for ambient RNA contamination estimation and adjustment.

For decontamination, we considered as empty the droplets with less than 10 UMIs (Young and Behjati, 2020) and estimated the fraction of background expression from each gene $g$ of library $j$ $(j = 1, \dots, 54)$ across all empty droplets $E$ as $b_g^{(j)} = \sum_{e=1}^{E} n_{g,e}^{(j)} / \{\sum_{e=1}^{E} \sum_{g=1}^{G} n_{g,e}^{(j)}\}$ where $n_{g,e}$ denotes the observed counts for gene $g$ in the empty droplet $e$. We used the background to estimate, likewise each cell's $c$ contamination fraction as $\rho_c^{(j)} = \{\sum_{g=1}^{G} n_{g,c}^{(j)} / \sum_{g=1}^{G} n_{g,c}^{(j)} \cdot \sum_{g=1}^{G} b_g^{(j)}\}$, where the sums are taken across all genes $G$ in each cell $c$ of library $j$ (SoupX's *autoEstCont* method). Finally, the endogenous (decontaminated) counts were retrieved as $\tilde{n}_{g,c}^{(j)} = n_{g,c}^{(j)} - \sum_{g=1}^{G} n_{g,c}^{(j)} \cdot \rho_c^{(j)} \cdot b_g^{(j)}$. The decontaminated counts of all genes $G$ in cells of library $j$ were stored in Seurat objects and were used for the downstream analysis. To assess the degree of contamination across all cells $N^*$, we estimated for each gene $g$, $g = 1, \dots, 36,601$ in library $j$, $j = 1, \dots, 54$, the difference in UMI counts before and after decontamination as $M_g = \frac{\sum_{j=1}^{J=54} [\overline{U}_g^j - \overline{C}_g^j]}{54}$, where $\overline{U}_g^j$ was the average uncorrected UMI of gene $g$ across all cells of library $j$ and $\overline{C}_g^j$ was the average corrected UMI of gene $g$ across all cells of library $j$. The $M_g$ levels were examined as a function of the average expression quantity $A_g = \frac{\sum_{j=1}^{J=54} [\overline{U}_g^j + \overline{C}_g^j]}{54}$ by a classical MA plot that indicated at each $A_g$ level the degree of average decontamination for gene $g$ across all libraries. In addition, we estimated the average contamination ranking for each $g$ as $r_g = \frac{\sum_{j=1}^{J=54} rank(M_g^j)}{54}$, where $M_g^j = \overline{U}_g^j - \overline{C}_g^j$. Combining the information, our data separated three broad clusters of contaminants, the most prominent of which included famous endocrine and exocrine markers such as Insulin (*INS*), Glucagon (*GCG*), Somatostatin (*SST*), Pancreatic Polypeptide (*PPY*), Regenerating Family Member 1 Alpha (*REG1A*), Serine Protease 2 (*PRSS2*) and other genes such as Transthyretin (*TTR*) and Islet Amyloid Polypeptide (*IAPP*). The genes in this cluster were ranked on average across all cells among the top 10 contaminants. A second cluster consisted of several mitochondrial and ribosomal genes ranked on average among the top 20–100 contaminants.

## Sample deconvolution by Demuxlet

We utilized modern barcoding technology to improve the throughput of detected cells and genes via genetic multiplexing (see "Genetic multiplexing") for a limited set of 12 libraries generated under the V3 chemistry. Each library consisted of multiplexed barcoded cells from two islets of donors with different clinical and demographic background and processed with Demuxlet (Kang et al, 2018). Demuxlet considered the islet's genetic variation to determine the genetic identity of each droplet through a set of single-nucleotide polymorphisms (SNPs). The islet SNPs were identified from the islet genotypes after extended quality control. The sample IDs of the processed genotypes were validated by comparing them to paired bulk ATAC-seq data using verifyBamID (Jun et al, 2012) (https://github.com/statgen/verifyBamID/releases). To obtain the SNPs, we used plink v1.90 (Purcell et al, 2007) and generated the Extended variant information files (.bim) accompanying the binary genotype information for each chromosome of the GRCh37 human genome. We performed error-correction for each file with HRC-1000G-check-bim (https://www.well.ox.ac.uk/~wrayner/tools/) to remove duplicate variants, mismatched variants, palindromic variants with frequency >0.4, and to correct strand flips using the reference file PASS.Variantsbravo-dbsnp-all.tab containing 170 M variants on ~15k individuals from the dbSNP database (Sherry et al, 1999). We joined and sorted the error-free data of all chromosomes with bcftools-1.11 (https://samtools.github.io/bcftools/bcftools.html) and, at the last step, we used liftOver to convert the genotype coordinates in the .vcf file to GRCh38 and obtained the barcode-to-islet associations.

Demuxlet quantified the likelihood that the $c^{(j)}$th droplet of library $j$ originated from the $m_1$ or the $m_2$ islets that have been multiplexed with mixing proportions $(1 - a) : a$. The likelihood

had the form:

$$L_d(m_1, m_2, a) = \prod_{v=1}^{V} \left[ \sum_{g_1, g_2} \left\{ \prod_{i=1}^{u_{c^{(j)}v}} \left( \sum_{e=0}^{1} (1-a) P(b_{c^{(j)}vi}, |, g_1, l) \right. \right. \right.$$
$$\left. \left. \left. + a P(b_{c^{(j)}vi}, |, g_2, l) \right) P_{m_1 v}^{(g1)} P_{m_2 v}^{(g2)} \right\} \right].$$

The above expression considers the reads from $C^{(j)}$ barcoded cell-containing droplets of library $j$ multiplexed across two islets $m_1$ and $m_2$. The islet genotypes are available across $V$ exonic variants, $u_{c^{(j)}v}$ is the number of unique reads overlapping the $v^{th}$ variant of the $c^{(j)}$ droplet, $b_{c^{(j)}vi}$ is the variant overlapping base call from the $i^{th}$ unique read, $i = 1, \ldots, u_{c^{(j)}v}$ representaing reference (R), alternative (A) and other (O) alleles and $l$ is a latent variable indicating whether the base call is correct (0) or not (1).

We used directly Demuxlet's .best output files that summarized the best assignment of the $c^{(j)}$-th droplet of $j$ between two multiplexed islets $m_1$ and $m_2$. The provided information explicitly associates each $c^{(j)}$ either to a single islet (singlets) or to both (Demuxlet doublet) with high probability. We integrated this information of each library with the Seurat soupX corrected objects and filtered out the Demuxlet doublets from further analysis. Across the 12 libraries, we removed on average 1065 doublets (25%–75% IQR: 650–1319). We found that the number of Demuxlet doublets was weakly anti-correlated to the library processing date (Pearson's $rho = -0.57$, $P$ value = 0.053) and also correlated to the number of STARsolo cell-containing droplets (Pearson's $rho = 0.611$, $P$ value = 0.035). We kept 401,305 cells for further analysis across $j^* = 1, \ldots, 66$ libraries holding the demultiplexed data of $D = 48$ islets (some islets are represented more than once).

## Quality control by library

Quality control (QC) analysis was performed iteratively on the decontaminated raw counts of each $j^* = 1, \ldots, 66$ demultiplexed library. It consisted of the following steps:

1. Preliminary filtering: We filtered out all cells with $nFEA \leq 500$ or $nUMI \leq 1000$ or $pMT \geq 50\%$ as a first-pass data cleaning for the subsequent pre-processing steps. The identification of high-quality cells combined multi-step doublet estimation, stricter $nFEA$, $nUMI$, and $pMT$ cutoffs and statistical testing as shown below.
2. Doublet estimation: We used Scrublet (Wolock et al, 2019) and DoubletFinder (McGinnis et al, 2019) to estimate the neotypic doublets from each $j^*$ library. Scrublet was run in Python 3.6.15 on the Seurat-to-10x formatted data (function write10xCounts of DropletUtils R package) with an expected 10% doublet ratio. We visualized the doublet scores of the observed and simulated doublets in a histogram and inspected their bimodal distributions to set an appropriate cut-off that separates the doublets from the singlets. DoubletFinder operated on the normalized and clustered Seurat data of each $j^*$. Similar to Scrublet, it simulated doublets with a set of user-defined parameters: $p_N$ (proportion of generated artificial doublets), $p_K$ (the PC neighborhood size to compute each cell's proportion of artificial k nearest neighbors on a PCA, $p_{ANN}$), $nPC$ (the number of principal components (PCs)) and $nExp$ (a threshold to make the singlet/doublet prediction, similar to the expected doublet ratio).

3. Clustering: We followed Seurat's v4.0 standard pre-processing workflow (Kang et al, 2018) from data normalization to cell clustering (see "Seurat clustering by library") to filter out low-quality cells before the main data integration step. We visualized the distribution of $nFEA$, $nUMI$, and $pMT$ of all cells across each cluster $x$ of $j^*$ to determine the appropriate cutoffs.
4. Marker analysis: To avoid over-filtering, we utilized marker expression analysis and pre-annotated the clusters with known endocrine and exocrine marker genes (see "Cell annotation"), considering that $nFEA$, $nUMI$, and $pMT$ may vary substantially across the various cell types.
5. $pMT$ comparisons: We examined whether certain cell types and, more importantly, glycemic states (ND, PD, T2D) exhibited higher $pMT$ rates than others to adjust the cutoffs. The comparisons were performed across annotated clusters within each state and across states within each cell type using ANOVA and Tukey's Honestly Significant Differences (HSD) pairwise tests with Bonferroni corrected $P$ values.

Scrublet set an automatic threshold at the point between the two modes of the simulated scores. We visually determined that the optimal doublet threshold was at 0.25 in all libraries. For DoubletFinder, we normalized and clustered the Seurat data (10 PCs, Leuven clustering with resolution = 0.5 on the UMAP) and instructed the algorithm to simulate doublets using $p_N = 0.25$ (default), $p_K = 0.09$ (estimated), $nExp = 0.1$ and $nPC = 10$ (default). Marker analysis showed that, in contrast to Scrublet, DoubletFinder often assigned higher doublet rates in $PPY^+$, $PECAM1^+$, and $COL1A1^+$ cell clusters. To avoid over-filtering, we removed only the common Scrublet-DoubletFinder doublets leaving a total of 336,692 cells for further processing (median = 6663; 25%–75% IQR = 4965–8288).

## Seurat clustering by library

Raw counts of each library were normalized with the *LogNormalize* model that employed a global-scaling (library size) normalization with *scale.factor* = 10,000 and log-transformed the result. The top 2000 variable features, exhibiting the highest cell-to-cell variation, were extracted, and their normalized counts were scaled to fit on a Principal Components Analysis (PCA) model for linear dimensionality reduction. We used standard quality control on the PC loadings to empirically determine the optimal number of PCs accounting for the data variability. For each library, we visualized the PC loadings and the associated heatmaps of gene expression and kept the first 100 PCs for the UMAP representation. We constructed a shared nearest neighbor graph by calculating the neighborhood overlap (Jaccard index) between every cell and its 20 nearest neighbors obtained from the cell Euclidean distances. We clustered the data with Leuven and a clustering resolution parameter equal to 1. We inspected separate violin and 2 d plots of $nFEA$ vs $pMT$ and $nUMI$ vs $pMT$. Based on the spatial 1 d and 2 d patterns, we flagged all cells with $nFEA < 1400$ unless, at these cutoffs, the flagged cells enriched for an annotated cluster (high-quality exocrine cell types had relatively low number of features) in which case, we determined that $nFEA < 1000$ was the most appropriate cut-off.

To set the $pMT$ cut-off, we compared various quantiles of the $pMT$ distribution across annotated cell types (derived by merging

the cells of the above annotated clusters) and glycemic states (ND, PD, T2D). We used ANOVA and Tukey HSD pairwise tests to find whether the $pMT$ rates differed significantly across the cell types. The analysis showed that α- and β-cells exhibited statistically higher $pMT$ medians, 70th quantiles, and 90th quantiles compared to other types (see "Quality control by library") at a Bonferroni-adjusted $P$ value 5%. On the other hand, none of the cell types showed significant differences among the $pMT$s of ND vs PD vs T2D at a Bonferroni-adjusted $P$ value 5%. Based on these plots and results, we flagged all cells with $pMT > 40\%$. All flagged cells were removed from further analysis, reducing the dataset to $\sum_{j^*=1}^{66} s^{(j^*)} = 257{,}070$ cells (median: 5135; 25%–75% IQR: 4063–6014).

## Data integration and final cell filtering

We used Harmony v3.8 (Korsunsky et al, 2019) within the Seurat v4.0 workflow to integrate the $j^* = 1, \ldots, 66$ libraries each consisting of $c^{j^*}$ cells (after quality control by library). Harmony exhibits excellent scaling properties for large populations and accommodates complex experimental designs, allowing the user to explicitly specify the model parameters (factors) to be integrated. First, we merged the normalized data across all 66 libraries (see "Seurat Clustering by library") and extracted the top 2000 variable features for scaling. The cells were embedded in a 100-dimensional PCA space and Harmony adjusted iteratively for sex, chemistry, and ancestry until convergence (10 iterations). At each iteration, the method used fuzzy clustering to assign each cell to multiple clusters while preserving the data diversity via the $\Omega(.)$ term that penalized statistical dependence between batch identity and cluster assignment. The estimated cluster and batch centroids were used to derive a batch correction factor and a cell-specific correction factor, which were iteratively updated until convergence to a stable clustering representation. The procedure generated 36 cell clusters. The median ratio of females across the 36 clusters was 0.36 (25%–75% IQR: 0.3–0.41), which was close to the overall female:male ratio of 0.37:0.63. Similarly, the median values were 0.81 for V3 chemistry (overall: 0.79), 0.49 for the European ancestry group (overall: 0.47), and 0.34 for the Hispanic ancestry group (median: 0.37).

## Doublet enrichment

We estimated which of the Harmony integrated clusters were enriched in Scrublet doublets by the Fisher test. For each integrated cluster $x^*$, $x^* = 0, 1, \ldots, 35$, we calculated the doublet ratios $DR_{x^*} = \frac{\text{\# of doublets in } x^*}{\text{\# of singlets in } x^*}$ and $DR_{-x^*} = \frac{\text{sum of doublets not in } x^*}{\text{sum of singlets not in } x^*}$. The enrichment test was performed on the $2x2$ confusion matrix with column entries the numerator and denominator of each of the above quantities. The clusters with $OR = DR_{x^*} : DR_{-x^*} > 2$ and Fisher test $FDR \leq 1\%$ were those labeled as significantly enriched. Clusters 29 with $DR_{29} = 1.248$ and 20 with $DR_{20} = 0.307$ stood out with relatively high ratios. The Fisher enrichment test showed that clusters 29, 20, 15, and 23 were significantly enriched in doublets (odds ratio, $OR > 2$; Fisher test FDR < 1%). We ran differential expression analysis at the single-cell level with Seurat's logistic regression model comparing the average expression of gene $g$ in cluster $x^*$ against its average expression in all other clusters after adjusting for sex, chemistry, and ancestry ($\log FC \geq 0.25$ and $FDR \leq 1\%$). Combined with evidence from marker expression

analysis, we found that cluster 29 expressed highly both $INS$ and $SST$ while cluster 20 expressed both $GCG$ and $SST$. None of the other clusters showed such evidence. We labeled all cells of 20 and 29 as doublets and filtered them out along with all other Scrublet doublets, ending up with the final set of $N = 245{,}878$ high-quality cells for downstream processing. We summarized the number of cells after each filtering step by islet and examined whether our strategy and cutoffs led to differences in cell percentages of each step across the glycemic states. We used ANOVA and Tukey's HSD to test the null hypothesis $H_0 : \bar{s}^i_{ND} = \bar{s}^i_{PD} = \bar{s}^i_{T2D}$ versus the alternative that at least one of the $\bar{s}^i$ differed, where $\bar{s}^i_{ND}$, $\bar{s}^i_{PD}$ and $\bar{s}^i_{T2D}$ were the average percentage of cells across the ND, PD, and T2D islets, respectively, after the filtering step $i$. We did not detect any statistically differences at Bonferroni-adjusted $P$ value = 5%.

## Doublet estimation

We estimated three types of doublets from our data. First, for the genetically multiplexed libraries, we found Demuxlet doublets consisting of cells from different donors (see "Sample deconvolution by Demuxlet"). Second, within each library, we utilized Scrublet and DoubletFinder that simulated doublets from the raw UMI counts (after Demuxlet when applicable) and, via nearest neighboring clustering, calculated the likelihood for an observed cell to be a doublet ("Quality control by library"). Third, after data integration, we identified clusters enriched in Scrublet doublets and expressing more than one of the known endocrine and exocrine markers. All cells of such clusters were potential doublets ("Doublet enrichment").

## Cell annotation

We annotated the integrated UMAP clusters of high-quality cells with a large list of known endocrine and exocrine marker genes obtained from the literature and by differential expression analysis. The latter was conducted by Seurat's logistic regression model that compared gene's $g$ average expression in $x^*$ against its average expression in all other clusters after adjusting for age, sex, BMI, ethnicity, and sequencing chemistry. We reported significant genes at $\log 2FC \geq 0.25$ and $FDR \leq 1\%$.

The estimated cell types (and indicative markers) were beta ($INS$), alpha ($GCG$), delta ($SST$), gamma ($PPY$), epsilon ($GHRL$), ductal ($KRT19$), acinar ($REG1B$), stellate ($COL1A1$), quiescent stellate ($FABP4$), endothelial ($PLVAP$), Schwann ($NGFR$), immune ($C1QC$), mast ($TPSB2$) and proliferating cells ($TOP2A$) (Appendix Fig. S3b,c). For each islet, we calculated the number of cells across the glycemic states of each cell type. We tested whether the absolute frequencies and/or the cell percentages differed significantly in ND vs PD vs T2D by ANOVA and Tukey's HSD pairwise tests with Bonferroni correction. Beta cells with $pMT \geq 40\%$, and other cell types with $pMT \geq 20\%$ were excluded from downstream analysis.

## Conversion to pseudo-bulk

We considered that the single cells within an islet are not independent of each other and estimated the transcriptomic differences across cell types and glycemic states at the pseudo-bulk level, where the islets served as the biological replicates. We aggregated the single-cell raw counts and metadata (clinical and

demographic information) associated with each cell type $z$ and islet $p$, $p = 1, \dots, P$, and generated $Z = 14$ pseudobulk RNA-seq count matrices of dimension $G \times P_z$, where $P_z$ might differ across the cell types. The aggregated data were obtained by the *aggregate.Matrix* function of the *Matrix.util* v0.9.8 R package. The quality control of the pseudobulk data was done in terms of the islet library sizes (*nUMI*), the islet number of detected genes (*nFEA*), the percentage of reads mapped to the MT genome (*pMT*), and the Counts-per-Million (CPM)-normalized log2-expression profiles of the top 50 most expressed genes. We employed Principal Components Analysis (PCA) on the top 2000 variable genes in edgeR v3.34.1 (Robinson et al, 2010) to evaluate the overall transcriptional differences across disease states, using the CPM-normalized and adjusted counts for sequencing chemistry, sex, ethnicity, scaled age, and scaled BMI. We utilized the variance partition mixed-effects model of the variancePartition R package to assess the major sources of gene variability attributed to the observed variables (disease state, sequencing chemistry, sex, ethnicity, age, and BMI), as well as the influence of unobserved factors measured by the model residuals.

## Cell type subclustering

The single-cell RNA-seq gene expression profiles of Alpha, Beta and Delta cell types were extracted and individually re-integrated as previously (see paragraph "Data integration"). The UMAP was estimated from the first 20 PCs and the clustering was performed with the Leuven method and a resolution parameter equal to 0.8. For each cell type, we compared gene's $g$ average expression in cluster $x_z$ vs all other clusters by Seurat's Negative Binomial (*negbinom*) model, adjusting for age, sex, ethnicity, and sequencing chemistry. We labeled genes as significant if $\log 2FC \geq 0.25$ and $FDR \leq 1\%$.

## Differential expression analysis with edgeR

We quantified the transcriptomic differences of the cell types and the glycemic states of each endocrine cell type by edgeR on the pseudobulk level. Each model was adjusted for sequencing chemistry, sex, ethnicity, scaled age, and scaled BMI. Cell type comparisons were done in the form of (1) cell type $z$ vs all other cell types to capture the global differences at $\log 2FC \geq 0.585$ and $FDR \leq 5\%$ and (2) all pairwise comparisons among the endocrine cell types to detect genes that were uniquely upregulated in $z$ at $|\log 2FC| \geq 1$ and $FDR \leq 5\%$. For glycemic states, we estimated all pairwise comparisons and reported differentially expressed genes at $|\log 2FC| \geq 0.585$ and $FDR \leq 5\%$. The functional characterization of the significant genes was done in *clusterProfiler v4.0* (Yu et al, 2012) using the *enrichGO* function with background all genes of the GRCh38 genome. The enriched biological processes with set sizes in $[10, 500]$ were reported at FDR = 5%.

## Enrichment of biological processes

Functional enrichment of the T2D vs. ND β-cell DEGs was conducted using Molecular Signatures Database (MSigDB, https://www.gsea-msigdb.org/gsea/msigdb/human/annotate.jsp). A false discovery rate (FDR) threshold of 0.05 was applied to identify significantly enriched biological processes. The top five non-

redundant processes for both upregulated and downregulated DEGs were visualized using a bubble plot.

## Weighted gene co-expression network analysis (WGCNA)

We estimated co-expressed gene modules from the pseudobulk β-cell data with the WGCNA R package (Langfelder and Horvath, 2008). WGCNA's input was the CPM-normalized and log2-transformed raw counts, adjusted for scaled age, sex, scaled BMI, ancestry differences, and sequencing chemistry with limma's removeBatchEffect model (Ritchie et al, 2015). Only the genes expressed in more than 20 islets were processed. From a wide range of soft thresholds $t \in \{1, \dots, 20\}$, we picked $t = 12$ that best satisfied the signed network's scale-free topology assumption and its trade-off with mean connectivity. We generated the Topological Overlap Matrix (TOM) and performed the module detection by average linkage hierarchical clustering on the TOM-based dissimilarities, setting minModuleSize = 100 and mergedCutHeight = 0.25, which returned 24 gene modules for downstream analysis. For module-trait association analysis, we calculated the Spearman nonparametric correlation coefficient $C_{m,j} = corr(eigen_m, trait_j)$ between each module's m eigengene $eigen_m$ and each trait $j$, and reported significant module-trait pairs at significance level $\alpha = 5\%$. Functional enrichment of the modules that were significantly correlated with T2D traits was performed with gProfiler's (Kolberg et al, 2020) gene over-representation test (https://biit.cs.ut.ee/gprofiler/gost) reporting significant hits at FDR ≤ 5%. Enrichment was defined based on the presence of ≥5 overlapping genes from the module within a given pathway.

## Functional annotation of DEGs

We obtained the summary statistics of all genetic variants significantly associated with T2D at genome-wide significance $P < 5 \times 10^{-8}$ from multiple ancestry metanalyses - T2DGGI (Suzuki et al, 2024), DIAMANTE (Mahajan et al, 2022), MVP (Vujkovic et al, 2020), and AGEN (Spracklen et al, 2020). For each T2D-associated variant, we identified proxy variants linked at $r^2 \geq 0.80$ across all ancestries based on 1000Genomes Phase 3 data. To determine if these T2D variants serve as eQTLs for our identified DEGs, we downloaded summary statistics data of *cis*-eQTL associations from TIGER (Alonso *et al*, 2021) atlas (https://tiger.bsc.es/), which includes 404 human pancreatic islet samples of European descent and reports >1.11 million significant eQTLs in >21,115 eGenes at 5% FDR. We retrieved all variants reported as eQTLs for our identified DEGs at $P$ value < 0.05, with a consistent direction of association across all four independent cohorts (+++ + or ----) in the TIGER dataset. We then cross-referenced the T2D genetic variant list and the eQTL list and identified eQTL variants for 41/511 DEGs that were also associated with T2D genetic risk.

Protein-level association data in T2D vs. ND whole islets were downloaded from the HumanIslets (Ewald et al, 2024) consortium (https://www.humanislets.com/#/) that reports mass spectrometry (MS)-based bulk protein expression data in 300 hand-picked islets per sample. $P$ values were calculated by multiple linear regression and adjusted for multiple testing with FDR.

Phenotypic data from whole gene knockout mouse lines were obtained from the International Mouse Phenotyping Consortium

(Groza et al, 2023) web portal (https://www.mousephenotype.org/). The latest data release (Nov 2024) reports 9073 phenotyped genes, with knockout mice produced and characterized by various institutional members of IMPC. All mice used in IMPC studies have a C57BL/6 N genetic background, with supporting mice derived from C57BL/6NJ, C57BL/6NTac, or C57BL/6NCrl strains. Glucose tolerance was assessed by initial response to glucose challenge and/or calculating the area under the glucose response curve from an intraperitoneal glucose tolerance test (IPGTT). Fat mass and lean mass were measured by Dual-energy X-ray Absorptiometry (DXA).

## Immunostaining and microscopy

Pancreatic islets from three ND donors (SAMN09370567, HP18054-01, HP18304) and three T2D donors (SAMN12634037, SAMN11642375, HP19044-01T2D) of the cohort (Table EV1) were embedded in optimal cutting temperature (OCT) compound, cryosectioned at 8 μm thickness using a Zeiss cryostat, and mounted onto glass slides. Frozen sections were allowed to equilibrate to room temperature, circled with a hydrophobic barrier pen, and air-dried for 20 min. Sections were then fixed in 4% paraformaldehyde for 15 min, followed by three 5-min washes in Dulbecco's phosphate-buffered saline (DPBS) to remove residual OCT. Permeabilization was performed using 0.2% Triton X-100 in DPBS for 15 min at room temperature, followed by three 5-min washes with DPBS. Sections were blocked with 5% normal goat serum (NGS, cat. no. 31872, ThermoFisher Scientific) in a humidified chamber at room temperature for 90 min. After aspiration of the blocking solution, primary antibodies—rat anti-C-peptide (cat. no. GN-ID4, Developmental Studies Hybridoma Bank, dilution 1:100), rabbit anti-glucagon (GCG; cat. no. 2760S, Cell Signaling Technology, dilution 1:100), and mouse anti-CHRNA3 (cat. no. MA5-31685, ThermoFisher Scientific, dilution 1:200)—were diluted in antibody incubation buffer (0.1% Triton X-100, 1% NGS in DPBS) and applied to the sections. Slides were incubated overnight at 4 °C in a humidified chamber. The following day, sections were washed three times in DPBS (5 min each) and incubated for 90 min at room temperature with secondary antibodies—goat anti-rat Alexa Fluor 568 (cat. no. A-11077, ThermoFisher Scientific), goat anti-rabbit Alexa Fluor 532 (cat. no. A-11009, ThermoFisher Scientific), and goat anti-mouse Alexa Fluor 647 (cat. no. A-21236, ThermoFisher Scientific)—each diluted 1:1000 in the same antibody incubation buffer. After secondary antibody incubation, sections were washed three times with DPBS and counterstained with DAPI-containing mounting medium (cat. no. P36966, ThermoFisher Scientific) for imaging.

Imaging was performed on a Leica TCS SP8 X DMi8 confocal microscope equipped with an ORCA-Flash4.0 digital camera (Hamamatsu) using a ×40 objective. Z-stack images were captured to visualize the full depth of immunostained islets. All imaging parameters—including laser power, detector gain, and exposure settings—were kept constant between ND and T2D islet samples. ND and T2D islet sections were processed in parallel and imaged the same day to minimize batch effects. Images were analyzed using CellProfiler (version 4.2.8) (Stirling et al, 2021). DAPI staining was used to segment nuclei, and these segments were expanded to approximate and capture whole cell boundaries. Donor-specific mean fluorescence intensities were used to classify β- (C-peptide[+])

and α-cells (GCG[+]), excluding unresolved (apparent double-positive) cells. CHRNA3[+] cells were subsequently identified within the resolved populations based on donor-specific mean intensity threshold. Islets from 3 ND and 3 T2D donors were analyzed; for each donor, three cryosections were imaged. The full CellProfiler analysis pipeline and parameters are provided in the Supplementary Materials.

## EndoC-βH3 culture

EndoC-βH3 cells were purchased from Human Cell Design (Toulouse, France) and cultured in Advanced DMEM F-12 media (Invitrogen) containing BSA (SIGMA), Glutamax (Gibco), 2-beta mercaptoethanol (SIGMA), nicotinamide (SIGMA), sodium selenite (SIGMA), Penicillin/Streptomycin (Gibco) and Puromycin (Calbiochem) on ECM (SIGMA) and Fibronectin (SIGMA) coated flasks.

## Lentivirus production and transduction of cells

Plasmid pLKO-puro shRNA clones (Mission shRNA) were purchased from SIGMA. Lentivirus was produced in HEK293T cells co-expressing the shRNA plasmid together with psPAX2 packaging plasmid and pVSVG envelope plasmid. Virus was concentrated using Lenti-X Concentrator (Takara) and virus titer was quantified using p24 ELISA antigen assay (Takara).

A MOI titer of 5 was used to transduce EndoC-βH3 cells at $1 \times 10^6$ cells in culture media without puromycin. Media change to puromycin complete media was done 18 h post transduction.

## RNA isolation

Total RNA was isolated from $3.5 \times 10^5$ cells/sample 96 h post transduction. Cells were collected for RNA extraction using TRIZOL (Invitrogen), phase separation was achieved using chloroform. Isopropanol was used for RNA precipitation using glycogen as a carrier; the pellets were washed using 75% ethanol, air-dried, and resuspended in DEPC water. RNA was measured using Nanodrop. Total RNA was used to perform qPCR using RNA to CT kit (Invitrogen) and FAM-Taqman probes (Invitrogen) and analyzed on QuantStudio 7 (Applied Biosystems) normalized to *ACTNB* or *TBP* Taqman probe.

## Insulin secretion assay

EndoC-βH3 cells infected with the lentivirus were seeded onto coated 24-well plates at $1.75 \times 10^5$ cells/well. Seventy-two hours post transduction, the cells were incubated overnight in Starvation media [DMEM no glucose, containing BSA (SIGMA), human Transferrin (SIGMA), Glutamax (Gibco), 2-beta mercaptoethanol (SIGMA), nicotinamide (SIGMA), sodium selenite (SIGMA)]. After 18 h, cells were equilibrated in KRBH buffer containing no glucose for 1 h, before stimulated insulin secretion was measured by static incubation of KRBH buffer containing 0 mM (Perez-Serna et al, 2023; Blanchi et al, 2023; Frørup et al, 2023; Szczerbinska et al, 2022) (Fig. 2) or 0.25 mM (Fig. 4) glucose and 20 mM glucose for 1 h. The supernatant was collected and stored at −20 °C until human insulin ELISA (Mercodia). After glucose stimulation, KRBH buffer was collected, and the cells were lysed with TETG solution [1 M Tris pH 8.0, Triton X-100, Glycerol, 5 M NaCl, 0.2 M EGTA,

and distilled water along with 1× cOmplete, protease inhibitor cocktail (Roche)]. The lysate was centrifuged at 3000 rpm for 5 min and stored at -20 °C until human insulin ELISA (Mercodia) according to the manufacturer's instructions. Total protein was measured using BCA kit (Thermo Fisher), and insulin secretion and content normalized to total protein content per sample.

## Apoptosis analysis by flow cytometry

EndoC-βH3 cells infected with the lentivirus were seeded onto coated 24-well plates at $1.75 \times 10^5$ cells/well. Ninety hours post transduction, cells were collected from the plate using Trypsin (Gibco) and stained using FITC Annexin V Apoptosis Detection Kit with 7-AAD (BioLegend) according to the manufacturer's instructions. The samples were run on Fortessa (BD Sciences), and data were analyzed via FlowJo Software (BD Sciences).

## Pseudoislets

Pseudoislets were prepared as previously described (Li et al, 2025; Walker et al, 2020). Briefly, to create β-cell-enriched human pseudoislets, primary human islets (donor information provided in Extended Data File 1) were cultured for 24 h in Prodo media, then aliquoted and washed with Versene (Gibco), dissociated with TrypLE (Gibco) for 2–3 min prior to incubation with mouse monoclonal anti-human NTPDase3 antibody (Ectopeptidases) for 30 min at 4 °C. NTPDase3-bound cells were washed in magnetic-activated cell sorting buffer (MB) containing 1× sterile phosphate-buffered saline (PBS; Corning), 0.5% FBS (Gemini), and 1% Penicillin-streptomycin solution (Gibco), then pelleted prior to resuspension and 15 min' incubation at 4 °C in MB containing anti-mouse IgG2a+b MicroBeads (Miltenyi Biotec). NTPDase3+ β-cells were purified using MS columns (Miltenyi Biotec), counted, and incubated with shRNA-packaged lentivirus (MOI = 5) for 2 h at 5% $CO_2$ 37 °C. Transduced cells (2,000 cells per well) were seeded in CellCarrier Spheroid Ultra-low attachment microplates (PerkinElmer) in enriched pseudoislet media as described (Walker et al, 2020). Cells were allowed to reaggregate for 5 days before live imaging on Opera Phenix Plus. shRNA target gene knockdown was confirmed using Taqman assay as described for EndoC-βH3 cells.

## Live-cell human β-enriched pseudoislet imaging

Live-cell imaging of human pseudoislet was completed using the Opera Phenix high-content screening system (Revvity) 60 min after labeling with the ReadyProbes™ Cell Viability Imaging Kit (Life Technologies) using DAPI (Hoechst 33342) and TRITC (PI) Laser/filter pairs. To obtain the whole spheroid/pseudoislet, we captured images using the 10x air objective in confocal mode (ROI 2160 × 2160) with only one field per well and 40 z-steps (5 μm step size) in 96-well CellCarrier Spheroid Ultra-low attachment microplates (PerkinElmer) maintained under cell culture conditions (37 °C, 5% $CO_2$).

## Image processing and analysis of β-enriched pseudoislet Hoechst-Propidium iodide staining

Automated image analysis was performed using Harmony High-Content Imaging and Analysis software (v5.1, Revvity). A pipeline was created to segment nuclei and measure cell numbers. Nuclei were identified based on Hoechst staining using z-stack processing set to 'max projection' with brightfield (but no flatfield) correction. First, we filtered each image in the DAPI channel using "smoothing Gaussian (width 2px)" and inverted the image (cut-off quartile = 100). Next, we assessed the full image ('ROI=none' setting) to identify distinct objects in the image with "absolute threshold" ≥220. In the resulting ROIs, we identified Hoechst-positive "nuclei" in the DAPI channel using method M (refer to Harmony software manual section 3.4). For each of these defined nuclei, we calculated propidium iodide staining intensity properties using the TRITC channel ('method=standard') and identified PI-positive cells with mean TRITC intensity >500 (Harmony software manual section 3.21). The number of $PI^+$ and $Hoechst^+$ cells based on these criteria was determined and exported using Harmony for each sample (average of 24 pseudoislets per shRNA per islet donor x 3 donors). Pseudoislet cell death for each shRNA target gene or NT shRNA control was calculated as percent $PI^+$/ $Hoechst^+$ cells for plotting, comparison, and statistical analysis.

## β-enriched pseudoislet volumetric analysis

Three-dimensional spheroid volume analyses were completed using imaging parameters described above with the following adjustments: 20 z-steps (10 μm step size) at brightfield (40 ms, 50% power). Analysis was done as 3D-analyses with brightfield but no flatfield correction. First, the brightfield image was adjusted using "smoothing Gaussian (width 4px)" and then inverted ("cut-off quartile = 100"). Next, we assessed the full image ("ROI=none" setting) to identify each pseudoislet using the following settings: "absolute threshold" ≥2000; "closing=10 μm"; "fill plane-wise and join touching fragments (volume >6000 μm³)". We calculated pseudoislet volume in μm³ using the "volume per object" output in "morphology properties ("method=standard")" from the Harmony software. Spheroids/pseudoislet volumes were exported for each well, aggregated, and plotted for *GRAMD2B*-shRNA or NT-CTR.

## TUNEL staining

A subset of shRNA-pseudoislets was embedded in OCT, and 8 μm cryosections were prepared using a Zeiss microtome. We confirmed β-cell enrichment in pseudoislets using GCG and C-peptide immunostaining as described above. Terminal deoxy-nucleotidyl transferase dUTP nick end labeling (TUNEL) was assessed for shRNA-transduced human β-enriched pseudoislets using the Apoptag Plus In situ Apoptosis Detection Kit (SIGMA) following the manufacturer's protocol. DAPI (Molecular Probes) was used to counterstain nuclei. Images were captured using a Leica SP8 STED confocal microscope with ×40 oil immersion objective. Systematic identification and counting of TUNEL-positive islet cells in confocal images were completed using Cellprofiler (analysis pipeline is provided in the Supplementary materials).

## Pancreatic immunohistochemistry and β-cell area assessment

Whole pancreata from 17-week-old mice were fixed overnight in Bouin's solution, embedded in paraffin, and sectioned at 5 μm

thickness using a microtome. Sections were deparaffinized, rehydrated, and subjected to immunostaining with a mouse monoclonal anti-Insulin antibody (cat. no. ab8304, Abcam, dilution 1:1000). Slides were counterstained with Mayer's hematoxylin and eosin, then scanned using a Hamamatsu NanoZoomer slide scanner at ×20 magnification. Digital images were analyzed with a custom macro application developed for β-cell area quantification (Visiopharma Aps). Quantification for beta cell area was performed on 10 sections per pancreas from each of 2 mice per genotype for a total of 40 sections (20/genotype).

## Statistics

Demographic differences between donor groups were analyzed by Games–Howell post-hoc test. All EndoC-βH3 experimental data were plotted as mean ± s.e.m and significance was calculated using Student's $t$ test (two-tailed, paired). Data from β-enriched pseudoislets, including PI staining, TUNEL assays, and volumetric measurements, were visualized using violin plots, and corresponding $P$ values were calculated using a mixed-effects linear regression model, with donor ID included as a random effect. For mouse trait measures, $P$ values were obtained as reported by the International Mouse Phenotyping Consortium (IMPC). Body weight trajectories were analyzed using two-way ANOVA, and $P$ values for β-cell area quantification in mice were calculated using two-tailed Student's $t$ test with unequal variance. Statistical significance is denoted as follows: $*P < 0.05$; $**P < 0.001$.

## Data availability

All human islet sample and single-cell RNA-seq datasets have been deposited in the BioProject and Gene Expression Omnibus databases under accession numbers PRJNA913127 and GSE221156. The data have been processed in the R statistical package and the analytical pipeline including the detailed methodology, the code and the associated plots/tables are available at Zenodo under https://zenodo.org/records/14656366. The reader can use our code outlined in the Pipeline_html.Rmd and Pipeline_html.html files to replicate the results and to explore other aspects of our data discussed in detail in the manuscript. At the single-cell level, the processed data are available for interactive visualization by cellxgene at https://cellxgene.cziscience.com/collections/58e85c2f-d52e-4c19-8393-b854b84d516e. The dataset is divided into four instances, one referring to the data of all annotated cells and one for each of the major identified cell types, namely Beta, Alpha, and Delta cells. At the pseudobulk level, processed data are available for interactive visualization by the TAPIC Rshiny applet at https://thejacksonlaboratory.shinyapps.io/TAPIC_Stitzel_Lab/.

The source data of this paper are collected in the following database record: biostudies:S-SCDT-10_1038-S44318-026-00744-w.

## Peer review information

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

## Acknowledgements

This study was made possible by generous funding from the American Diabetes Association Pathway to Stop Diabetes Accelerator Award (1-18-ACE-015) and National Institutes of Health (NIH) award number R01DK118011 (to MLS) as well as Department of Defense Congressionally Directed Medical Research Program (CDMRP) award number W81XWH-18-0401 (to MLS and DU). CNS was also supported by American Diabetes Association grant 11-22-JDFPM-06. Opinions, interpretations, conclusions, and recommendations are solely the responsibility of the authors and do not necessarily represent the official views of ADA, NIH, or DOD. We gratefully acknowledge contributions of JAX Single Cell Biology, Genome Technologies, the Histology Core, Center for Biometric Analysis, and Microscopy services and Research Cyberinfrastructure computational resources at The Jackson Laboratory for expert assistance with the work described in this publication. We are indebted to the anonymous islet organ donors and their family, which were provided by the NIDDK-funded Integrated Islet Distribution Program (IIDP) (RRID:SCR_014387) at City of Hope (2UC4DK098085). This study used data from the Organ Procurement and Transplantation Network (OPTN) that was in part compiled from the data hub accessible to IIDP-affiliated investigators through the IIDP portal (https://iidp.coh.org/secure/isletavail). The OPTN data system includes data on all donors, wait-listed candidates, and transplant recipients in the US, submitted by the members of OPTN. The Health Resources and Services Administration of the US Department of Health and Human Services provides oversight of the activities of the OPTN contractor.

The data reported here have been supplied by UNOS as the contractor for OPTN. The interpretation and reporting of these data are the responsibility of the author(s) and in no way should be seen as an official policy of or interpretation by the OPTN or the US government. Special thanks to Dr. Raphael Scharfmann at the Institute Cochin for help optimizing EndoC-βH3 culture. We thank Ucar and Stitzel lab members for critical feedback throughout this study and Dr. Scott Soleimanpour for helpful advice during revisions.

## Author contributions

**Khushdeep Bandesh**: Data curation; Formal analysis; Validation; Investigation; Visualization; Writing—original draft; Writing—review and editing. **Efthymios Motakis**: Resources; Formal analysis; Investigation; Visualization; Methodology; Writing—original draft; Writing—review and editing. **Siddhi Nargund**: Resources; Data curation; Formal analysis; Validation; Investigation; Visualization; Writing—original draft; Writing—review and editing. **Romy Kursawe**: Conceptualization; Data curation; Formal analysis; Validation; Investigation; Visualization; Methodology; Writing—original draft; Writing—review and editing. **Vijay Selvam**: Formal analysis; Validation; Investigation. **Ansarullah**: Formal analysis; Validation; Investigation; Visualization; Methodology; Writing—original draft; Writing—review and editing. **Redwan M Bhuiyan**: Validation; Investigation. **Giray Naim Eryilmaz**: Resources; Data curation; Software; Validation; Investigation; Visualization. **Amelia M Willett**: Resources; Validation; Investigation; Methodology. **Jacqueline K White**: Resources; Validation; Investigation. **Sai Nivedita Krishnan**: Formal analysis; Validation; Investigation. **Cassandra N Spracklen**: Conceptualization; Resources; Formal analysis; Supervision; Writing—original draft; Writing—review and editing. **Duygu Ucar**: Conceptualization; Supervision; Funding acquisition; Writing—original draft; Project administration; Writing—review and editing. **Michael L Stitzel**: Conceptualization; Resources; Data curation; Formal analysis; Supervision; Funding acquisition; Investigation; Methodology; Writing—original draft; Writing—review and editing.

Source data underlying figure panels in this paper may have individual authorship assigned. Where available, figure panel/source data authorship is listed in the following database record: biostudies:S-SCDT-10_1038-S44318-026-00744-w.

## Disclosure and competing interests statement

The authors declare no competing interests.

