## [Peer Review File · The EMBO Journal]

Deep single-cell decoding of human pancreatic islets reveals T2D β -cell gene expression defects

Khushdeep Bandesh, Efthymios Motakis, Siddhi Nargund, Romy Kursawe, Vijay Selvam, . Ansarullah, Redwan Bhuiyan, Giray Eryilmaz, Amelia Willett, Jacqueline White, Sai Krishnan, Cassandra Spracklen, Duygu Ucar, and Michael Stitzel

Corresponding author: Michael Stitzel (michael.stitzel@jax.org)

Review Timeline:

Submission Date:	14th Jan 26
Editorial Decision:	15th Jan 26
Revision Received:	26th Jan 26
Editorial Decision:	30th Jan 26
Revision Received:	10th Feb 26
Accepted:	23rd Feb 26

Editor: Daniel Klimmeck

Transaction Report:

Dear Dr Stitzel,

Thank you for transferring your manuscript for consideration by the EMBO Journal. As discussed yesterday, please submit the current amended version of your manuscript files as well as a complete response towards the referees' second round concerns (preferably in doc format) using the link enclosed below. No reformatting is required at this stage.

Upon resubmission, I will swiftly proceed with the arbitration process as mentioned.

Thank you again for your interest in the EMBO Journal for your work.

Please let me know any time should there be additional questions related.

Kind regards,

Daniel Klimmeck

Daniel Klimmeck, PhD
Senior Editor
The EMBO Journal

Read our guidance for manuscript revisions and related editorial policies: <https://link.springer.com/journal/44318/submission-guidelines#cms-Revised-submissions>

<https://media.springernature.com/original/springer-cms/rest/v1/content/27825798/data/v1>

- a point-by-point response to the referees' comments, with a detailed description of the changes made (as a word file).
- a word file of the manuscript text.
- individual production quality figure files (one file per figure)
- a complete author checklist
- Expanded View files (replacing Supplementary Information)
- a Reagents and Tools Table as part of the Methods section

We realize that it is difficult to revise to a specific deadline. In the interest of protecting the conceptual advance provided by the work, we recommend a revision within 3 months (15th Apr 2026). Please discuss the revision progress ahead of this time with the editor if you require more time to complete the revisions.

Response to Reviewers:

We appreciate the reviewer requests for additional details on specific aspects of the methods and analytical pipeline, which we agree will further enhance transparency and reproducibility of this study for readers and future analyses. In response, we have expanded the relevant sections of the manuscript to provide greater methodological detail and facilitate replication of this dataset by others in the field. In addition to more detailed description in the Methods, we have revised sections of the Results and Discussion to better highlight unique aspects and deliverables of our study and better contextualize our approach and results relative to prior studies, highlighting how improved data quality in the present work enabled the identification of a more robust set of differentially expressed genes compared with earlier reports.

We also provide additional point-by-point responses to each reviewer's critiques and comments below:

Reviewer #1:

In the revised version, the authors have added functional validation for multiple T2D-associated genes, which strengthens the overall credibility of their conclusions. Nevertheless, several critical issues remain that need to be addressed.

1. Figure 1 still lacks sufficient novelty, as its main conclusions largely reiterate previously published findings (PMID: 37231096).

Response: In line with standard resource articles, Figure 1 summarizes the basic characteristics of the study cohort to inform readers about the central foundation supporting all subsequent analyses and experimental follow-up. This figure is intended to provide a factual description of the dataset and to place our observations in the context of previously reported studies (PMID: 37231096 and PMID: 34731614). To better highlight the low exocrine contamination of our single-cell transcriptomes and resulting atlas which is one of the factors likely contributing to our larger number of differentially expressed beta-cell genes compared to the latest HPAP cohort-based analyses (PMID:37231096), **revised Figure 1d** now shows the endocrine/exocrine/immune proportions obtained for each donor (**left panel**) and distribution of all cell types captured and identified (**right panel**), rather than endocrine-restricted proportions.

We have further revised the manuscript text, lines 119-122, to move our acknowledgement of previous reports from the end of the sentence in the previous manuscript version to a position at the beginning of the sentence:

“Consistent with previous reports^{13,14}, overall β -cell/endocrine proportions were 13-15% lower in T2D islets compared to ND or PD donor islets (**Figure 1e**; mean β -cell percentages: T2D=42.2±11.3; ND=55.2±10.7, p=0.006; PD=57.2±12.9, p=0.002, ANOVA followed by Tukey's honest significance test).”

To enhance novelty, the authors propose that the proportion of functional β cells is reduced by approximately 25–30% in T2D donors compared with ND or PD donors. However, this conclusion is not convincingly supported.

First, it remains unclear how “functional β cells” are defined. No direct functional assays are provided to demonstrate that β cells in cluster 1 indeed exhibit superior functional capacity. Moreover, if cluster 1 is considered the sole population representing functional β cells, does this imply that the majority of β cells in ND samples are non-functional? Clarification and stronger functional evidence are required to support this interpretation.

Response: In the previous manuscript revision, we de-emphasized the subpopulation analyses throughout the manuscript text and moved these results from the main figures to the supplementary data to reorient the focus on the functional genes identified in this study. Compared to other islet single-cell transcriptome studies, this study provides important systematic, unsupervised comparison of endocrine cell subpopulations across glycemic states and supports previous targeted studies (PMID: 27399229, PMID: 26389675, PMID: 27667365, PMID: 31500834, PMID: 27398620, PMID: 36928765, PMID: 22980982 and PMID: 27345837), therefore we provide the subpopulation data and analyses as supplementary material for researchers interested to investigate the functional roles of putative β -cell subpopulations which is beyond the scope of our study.

We have further removed the phrase “functional β cells” from the abstract (lines 26-28) and the results (lines 166-174) as follows:

Abstract, lines 26-28:

“This revealed ~25-30% β -cell reductions consisting of β -cell loss and proportional increases in a senescent β -cell subpopulation in T2D vs. ND or PD donors, consistent with previous reports.”

Results, lines 166-174:

“The cluster 1 subpopulation, with elevated expression of genes involved in insulin secretion, was reduced by an average of 10.5% in T2D vs. ND β -cells ($p = 0.001$). In contrast, the proportion of ‘cellular senescence’ cluster 7 cells increased by an average of 12.3% in T2D vs ND β -cells ($p = 0.009$); this significant increase was also observed in T2D vs. PD β -cells ($p = 0.02$, 9.7% average increase). These unsupervised subpopulation analyses thus support emerging reports of increased β -cell senescence in T2D^{27,28,29}. Together, these subpopulation shifts, combined with 10-15% overall reductions in T2D donor β -cell numbers/proportions in this cohort (**Figure 1e**), implicate ~25-30% reduction of β -cells in T2D vs. ND or PD donors.”

2. In Supplementary Fig. 4f, the authors report distinct proportions of ND- and T2D-derived β cells distributed between cluster 1 and cluster 7. However, performing Louvain clustering on data that do not show clear separation in the integrated UMAP space is methodologically questionable.

While genes associated with insulin secretion are more highly expressed in cluster 1 and genes related to cellular senescence are enriched in cluster 7, several key issues

remain unresolved. Did the authors observe an increased proportion of cluster 7 β cells in aging donors? Additionally, do the genes defining cluster 1 and cluster 7 overlap with the differentially expressed genes identified in Fig. 2a? Addressing these points would help clarify the biological relevance of these clusters.

Response: We did not observe significantly increased proportions of cluster 7 β -cells in aging donors (Reviewer Figure 1). 6% (19/ 316) of T2D upregulated DEGs overlapped with Cluster 7-defining genes, and 6.6% (13/195) of T2D downregulated genes overlapped with Cluster 1-defining genes. Importantly, no cluster 7- or cluster 1-defining genes overlapped with T2D downregulated or upregulated DEGs, respectively.

Reviewer Figure 1: Beta cell subcluster 7 proportions do not significantly increase with age (% cells in Cells in Cluster 7

cluster 7 (y-axis) vs. donor age (x-axis) in this study's 48 donor cohort).

3. The authors conclude that gene expression differences between PD and ND samples are minimal. However, the PCA plot referenced in Reviewer 2, comment #5 shows substantial inter-donor variability. Notably, several T2D samples cluster closely with ND and PD samples, occupying overlapping regions of the PCA space.

This raises concerns that the observed distributions may be influenced by confounding factors such as donor age, sex, BMI, or ethnicity. It is also unclear whether excessive inter-individual variability may be masking genuine transcriptional differences in the group-level comparison between PD and ND samples. Have the authors formally evaluated the contribution of donor-level heterogeneity to this analysis? Furthermore, does a similar issue of high inter-sample variability also affect the comparison between T2D and ND samples?

Response: The PCA plot referenced in Reviewer 2, comment #5 in our previous response letter was not adjusted for BMI. However, our differential gene expression analysis is adjusted for all covariates noted by the reviewer, including donor age, sex, BMI, ethnicity, and sequencing chemistry. We thank the reviewer for identifying this oversight. In the revised PCA plot (**Reviewer Figure 2**), adjusted for age, sex, BMI, ethnicity, and sequencing chemistry, almost all T2D samples (16/17) cluster distinctly from ND and PD samples along principal component 2 (PC2; y-axis), which accounts for 14% of the total variance and distinguishes T2D from ND and PD samples. Modest overlap along the periphery of the clusters is expected in human data and reflects the inherent biological variability rather than a lack of between-group differences.

Reviewer Figure 2: Principal component analysis of β -cell gene expression from ND, PD and T2D donors, adjusted for age, sex, BMI, ethnicity, and sequencing chemistry. Clusters are shown with 95% confidence intervals. 16/17 T2D samples cluster separately from ND and PD samples.

We notice inter-individual variability among our samples. However, all three groups (ND, PD and T2D) exhibit similar degree of sample variability, and the variability between the groups is lower than within each group.

Reviewer #2:

I thank the authors for the extensive effort invested in addressing my previous comments. The revision includes considerable additional experimentation and represents a technical improvement over the original submission. However, despite these efforts, the manuscript remains largely descriptive and does not provide a unifying biological mechanism or conceptual advance.

1. Lack of novelty

Although the manuscript is positioned as a Resource paper based on human islet scRNA-seq profiling, novelty remains a major unresolved concern. As raised in the previous round, the field already contains multiple larger-scale and more comprehensive single-cell and multiomic datasets of human islets across ND, PD, and T2D states. In response, the authors primarily reframe novelty as improvements in data quality and validation depth. However, the claimed higher-quality scRNA-seq alone does not constitute novelty, particularly in a field that is already saturated with comparable or larger datasets.

Although it is improved with more validation results, it remains unclear what new biology emerges from this dataset that could not have been inferred from previously published studies. The gene prioritization strategy is standard rather than conceptually novel, and the functional validation largely converges on β -cell viability and apoptosis, processes already well established in T2D β -cell dysfunction.

Overall, the revised manuscript represents another human islet scRNA-seq resource with functional add-ons, but it still lacks a clear conceptual advance

Response: Beyond improvements in single-cell RNA-seq data quality, our Resource study advances the field by integrating functional data across complementary *in vitro* and *in vivo* models to interrogate key T2D pathophysiological processes, including insulin secretion and β -cell viability. To highlight the novelty of our findings:

- We identified significant upregulation of neurotransmitter receptor genes involved in β -cell neurotransmission in T2D, a feature not apparent in prior studies. As proof of principle, we demonstrate increased protein expression of the nicotinic acetylcholine receptor CHRNA3 in T2D islet sections.
- We provide the first functional evidence that reduced expression of vitamin A metabolism genes compromises human β -cell survival.
- We demonstrate causal roles for ten newly identified T2D-associated downregulated genes in beta-cell death and dysfunction (*MPP1*, *CD82*, *GLUL*, *GOLT1A*, *RDH12*, *BCO1*, *ARG2*, *PITPNM2*, *PDZK1*, and *GRAMD2B*).
- Unlike earlier descriptive studies, we assessed the function of a T2D down-regulated gene, *GRAMD2B* using multi-system (human EndoC- β H3 cells, primary human pseudoislets, knockout mouse), multi-species (human and mouse), and multimodal (molecular measures of cell death, *in situ* histologic analysis of islet insulin content and beta-cell area, metabolic phenotyping) approaches to demonstrate its role in beta-cell death, which is a pathologic feature in islets associated with and contributing to T2D.

2. Descriptive nature and limited biological insight

While the authors have added candidate-based functional validation using knockdown approaches in beta-cell lines and human islets, these experiments largely stop at phenotype description (cell death or GSIS) without deeper mechanistic investigation. Across multiple sections, the study presents disconnected validation experiments without a coherent biological narrative.

While EndoC cells and pseudoislets are useful tools, most functional readouts converge on viability/apoptosis or GSIS, without elucidating signaling pathways, transcriptional regulation, or disease-driving mechanisms. As a result, the biological insight remains superficial, and the study does not clearly explain how or why these genes contribute to β -cell dysfunction in T2D.

Response: We respectfully assert that the experiments demonstrating impaired islet function and increased beta-cell death, which are phenotypic hallmarks of T2D, are centrally relevant outcomes that experimentally link T2D-associated gene expression alterations to pathologic outcomes and motivate future mechanistic studies. Our objective was to bridge high-quality human single-cell discovery with causal functional relevance in human β -cells, addressing a recurrent gap in the field where scRNA-seq-derived signals remain unvalidated. While detailed studies of downstream mechanisms are beyond the scope of this Resource manuscript, we provide pathway-level insight (e.g., neurotransmitter signaling and vitamin A metabolism) and validation in multiple experimental models (human β -cell line, primary pseudoislets, and *in vivo* mouse model), which we believe is a significant contribution to the field.

In view of these contributions, we have revised lines 544-546 in the Discussion section to clarify this scope:

“This study begins to translate human single-cell associations into experimentally validated, disease-relevant gene targets, which we hope will facilitate and inform future mechanistic work.”

3. Data interpretation and consistency concerns

Several aspects of the revised analysis raise concerns regarding data interpretation:

Stellate cell annotation (Supplementary Fig. 2c):

Active stellate cells are reported to express FABP4; however, FABP4 is a well-established marker of quiescent pancreatic stellate cells and has been reported as such in prior human islet scRNA-seq studies (e.g., Cell Systems 2016; Cell Reports 2019). This raises questions about cell-state annotation.

Response: We thank the reviewer for identifying this error. We have corrected the nomenclature throughout the manuscript and in the related deposited datasets, replacing “activated” stellate cells with “quiescent” stellate cells.

GRAMD2B expression inconsistency:

GRAMD2B is reported as downregulated in T2D β -cells in this study, whereas other

public datasets report the opposite trend. A systematic comparison across datasets would be necessary to reconcile this discrepancy. <http://tools.cmdga.org:3838/isletHPAP-expression/>

Response: We evaluated *GRAMD2B* β -cell expression in the dataset provided above by the reviewer, which also shows a trend of decreased *GRAMD2B* expression in T2D (TPM=2.065) vs. ND (TPM=2.758) beta-cells. A screenshot of the results from this tool is provided below:

Additionally, HumanIslets.com (PMID:39357523) indicates that *GRAMD2B* gene expression is lower in T2D islets than in ND (or pre-T2D) islets (Reviewer Figure 3), and data from Bacos K *et al.* 2023 (PMID: 36656641) also indicates *GRAMD2B* expression is significantly lower in T2D vs. ND donor islets (Mean \pm SD: T2D 435.2 \pm 89.1; ND 575.9 \pm 113.4, log₂FC=-0.33, p=1.5E-09). Thus, three independent studies support our discovery with a trend in beta-cells (CMDGA above) or significant (PMIDs:39357523, 36656641) down-regulation of *GRAMD2B* expression in T2D islets.

GRAMD2B: GRAM domain containing 2B

Reviewer Figure 3. *GRAMD2B* gene expression in whole islets from ND, T2D donors as reported in HumanIslets.com

Candidate gene selection:

The criteria for selecting genes for functional validation are unclear. For example, several genes validated in Fig. 2 are not prominently highlighted in the discovery analyses, yet knockdown of all selected genes produces impaired insulin secretion, raising concerns about selection bias and interpretability.

Response: For the initial functional validation (Figure 2), we randomly selected candidates from downregulated T2D DEGs with the lowest FDR (*MPP1*, *CD82*, *GLUL*, and *GOLT1A*), all of which were previously unreported in the context of T2D. *FXVD2*, a well-established T2D-associated β -cell DEG, was included as a positive control. Although knockdown of all four candidate genes affected insulin secretion, *MPP1* was the only gene whose knockdown increased insulin content, and *GLUL* was the only gene whose knockdown increased apoptosis.

Previously unreported genes (Suppl Fig. 8d):

The identification of ~340 genes not reported in prior scRNA-seq or sorted β -cell RNA-seq datasets raises questions regarding annotation, thresholds, or technical artifacts that should be addressed.

Response: As reported in Methods, lines 870-873, cells were annotated using standard marker genes: beta (*INS*), alpha (*GCG*), delta (*SST*), gamma (*PPY*), epsilon (*GHRL*), ductal (*KRT19*), acinar (*REG1B*), stellate (*COL1A1*), quiescent stellate (*FABP4*), endothelial (*PLVAP*), Schwann (*NGFR*), immune (*C1QC*), mast (*TPSB2*) and proliferating cells (*TOP2A*), consistent with well-documented studies, including our previous work (PMID: 27864352). Differential gene expression analyses were adjusted for potential confounders, including sequencing chemistry, sex, ethnicity, scaled age and scaled BMI (Methods, lines 907–908), and differentially expressed genes were defined using an FDR threshold $\leq 5\%$ and a \log_2 fold change ≥ 0.585 (Methods, lines 911-913).

4. Interpretation of PD samples

The modest separation between ND and PD samples and minimal differential expression should be clearly framed as a negative result to present. PD should be considered as ND in the analysis.

Response: We have included the modest separation between ND and PD samples and minimal differential expression in Results, lines 180-186:

“Principal component analysis (PCA) of β -cell transcriptomes suggested modest differences in transcriptional profiles of PD vs. ND donors (data not shown), but only one gene was differentially expressed at FDR $<5\%$ (PD vs. ND, **Supplementary Table 10**). These data suggest that, while PD may represent an early transitional state preceding major remodeling observed in T2D, it is represented by subtle expression changes. Alternatively, the data may

indicate that islets from PD donors are not significantly altered and that the PD state instead results from or reflects changes in donor insulin resistance.”

We disagree that PD donors should be considered as ND in the analyses. As described in line 77-79, donors were designated as PD based on American Diabetes Association (ADA)'s prediabetes criteria ($5.7\% \leq \text{HbA1c} \leq 6.4\%$). PD donors have higher glycosylated hemoglobin levels than ND donors, reflecting an intermediate glycemic state that is biologically and clinically distinct from normoglycemia.

Lines 77-79: “17 diagnosed T2D (mean HbA1c = 7.6%), 14 PD (mean HbA1c = 5.9%; designated based on American Diabetes Association (ADA) prediabetes criteria ($5.7\% \leq \text{HbA1c} \leq 6.4\%$))12), and 17 ND (mean HbA1c = 5.2%) donors”

5. Figure-specific conceptual concerns. For example:

Figure 2f:

The presentation of 24 β -cell gene modules is difficult to interpret, and the figure does not clearly convey a biological message or conceptual advance. The meaning of color-coded dots and the functional implications of these modules require clarification.

Response: The WGCNA analysis (Figure 2f) was completed in response to reviewer 1's comment in the previous revision to address concerns about more innovative analysis and identifying gene regulatory networks. The goal was to determine whether the disease-associated transcriptional changes identified by T2D DEG analysis converge into cohesive co-expression modules corresponding to biologically relevant pathways. Importantly, this analysis identified two modules—the 'black' and 'gray60' modules—that are significantly enriched for neuroreceptor genes and vitamin A metabolism genes, respectively, consistent with the pathways highlighted by our DEG analyses.

To improve the figure clarity, **revised Figure 2f** now includes legends for the color-coded dots in the plot: purple indicates upregulated genes, blue indicates downregulated genes, and dark gray represents the total number of genes within each module. The correlation-based color scale remains described in the scale key. We believe these changes improve the interpretability and conceptual clarity of **revised Figure 2f**.

Figure 3e–k (Vitamin A pathway):

The authors propose that downregulation of vitamin A (VA) metabolism–related genes in T2D leads to reduced VA availability, subsequent suppression of VA target genes (e.g., ARG2, NEDD9), and ultimately β -cell death. However, this model is largely speculative, and several key assumptions are not supported by the presented data or existing literature. 1) no direct evidence that β -cells produce less VA under T2D (experiment needed). 2) the predicted VA-related differentially expressed genes are not functionally tested for their ability to regulate VA production in β -cells and at T2D (experiment needed). 3), there is no established mechanistic link between vitamin A metabolism and ARG2 function. ARG2 is primarily involved in arginine metabolism, and while RXR/RAR signaling can broadly regulate transcription, this indirect relationship

does not establish ARG2 as a functional component or target of VA metabolism (experiment needed). It is also unclear whether ARG2 itself is a robust T2D DEG across datasets (?). 4) The downstream mechanism by which ARG2 would promote β -cell apoptosis is not explored. As currently presented, the proposed VA–ARG2–apoptosis axis lacks sufficient experimental support and should be interpreted with greater caution.

Response: We do not claim that β -cells produce less vitamin A in T2D. Instead, we show that key enzymes required for VA metabolism (RDH12 and BCO1) are downregulated in T2D β -cells, and that knockdown of these genes increases human β -cell apoptosis ~2-fold. This data supports a causal relevance for intact VA metabolic capacity in maintaining β -cell viability, independent of direct VA quantification. Measuring Vitamin A levels in T2D vs ND β -cells, testing the effect of target gene knockdown on Vitamin A production, experimentally validating ARG2 as Vitamin A target in β -cells, and investigating downstream mechanisms of ARG2 function would require extensive new studies that are beyond the scope of this study.

While our data suggests a functional association between genes of the vitamin A metabolism pathway and β -cell viability in T2D islets, we acknowledge that definitive causal relationships will require further investigation.

In light of these considerations, we have restructured text in the Results and Discussion sections to distinguish our results from others' models and to moderate our inference of vitamin A dysregulation in T2D:

Moved from Results to Discussion, lines 511-517: “Dietary vitamin A deficiency has been linked with hyperglycemia, mirroring reduced vitamin A levels in the pancreas⁶⁰. Pancreas β -cells generate 9cRA (an active vitamin A metabolite and a high affinity ligand for RXR)¹¹⁴, and β -cell mass reduction (e.g., heterozygous Akita and streptozotocin-treated mice models) is accompanied by proportional decreases in 9cRA levels⁶⁶. In β -cells, elevated glucose concentrations suppress 9cRA biosynthesis⁶⁴, a central process in vitamin A metabolism, governed by the reductases RDH10 and RDH12.”

Results, lines 300-309: “T2D β -cells exhibited differential expression of several genes encoding cell death-associated proteins⁶³, including reduction of ARG2 and NEDD9 and induction of FAIM2, NUPR1, GAS6, HGF, and RAMP3, none of which were altered in T2D α -cells (**Figure 3e**). Notably, ARG2 and NEDD9 promoters harbor response elements for RXRA (**Supplementary Figure 14**), which is directly activated by the key vitamin A metabolite 9-*cis* retinoic acid (9cRA)^{64,65}.”

To test if T2D down-regulated vitamin A metabolism pathway genes alter β -cell survival, we completed shRNA-mediated knockdown of BCO1 (a rate-limiting enzyme in vitamin A metabolism⁶⁶), RDH12 (involved in 9cRA biosynthesis)⁶⁷, and ARG2 (a downstream target of vitamin A metabolism) in human EndoC- β H3 β -cells (**Supplementary Figure 13b**).”

Results, lines 319-320: “Together, these results suggest that down-regulation of genes in the vitamin A metabolism pathway in T2D reduces β -cell viability in human islets.”

Figure 5 (GRAMD2B KO mouse):

The in vivo phenotype of the GRAMD2B knockout mice is not sufficiently anchored to β -cell-intrinsic dysfunction. Glucose tolerance measurements alone are difficult to interpret, as the observed phenotype could reflect effects from other organs or systemic metabolic alterations. Direct assessment of insulin secretion during glucose challenge, as well as ex vivo islet functional analyses, would be necessary to determine whether the defect is islet autonomous or secondary to developmental, immune, or extrapancreatic effects.

Response: Knockdown experiments in human β -cell line and primary human β -cell pseudoislets clearly inform that perturbation of *GRAMD2B* lowers β -cell survival, a β -cell intrinsic defect, resulting in smaller pseudoislets. These human β -cell derived functional data are consistent with the reduced islet size observed in *Gramd2b* KO mice.

In addition, the phenotype appears male-specific, yet this sexual dimorphism is not explored or discussed. The absence of glucose intolerance in female knockout mice raises important questions regarding islet morphology and function in females, which are not addressed. Mechanistic insight into how GRAMD2B loss leads to impaired glucose homeostasis is also lacking, limiting interpretation of causality.

Response: Lack of overt glucose intolerance in female knockout mice may reflect sex-dependent compensatory mechanisms, including differences in insulin sensitivity, hormonal regulation, or β -cell reserve. However, the detailed analyses of islet cellular composition and morphology in female knockout mice were not performed in the IMPC dataset. Therefore, we cannot confidently comment on these aspects at the time. Human β -cell and pseudoislet experiments indicate that *GRAMD2B* inhibition leads to elevated β -cell death. We look forward to mechanistic dissection of *GRAMD2B* and other T2D DEGs in future studies.

Finally, as the GRAMD2B knockout phenotyping relies on IMPC-generated data, greater methodological detail is needed to evaluate these results, including the age of mice, glucose tolerance testing protocol, and presentation of full glucose excursion curves. Without these details, it is difficult to assess the robustness and β -cell relevance of the reported phenotype.

Response: We appreciate the reviewer's request for additional methodological detail regarding the IMPC-derived GRAMD2B knockout phenotyping. Because these experiments were conducted as part of the standardized IMPC pipeline, it is not practical to incorporate the full methodological protocols within this manuscript. However, we now provide an additional IMPC citation in the manuscript that directs readers to the IMPC protocol repository providing comprehensive information on all tested protocols, phenotyping methods, and knockout mouse generation.

References Cited:

Meehan TF, Conte N, West DB, et al. Disease model discovery from 3,328 gene knockouts by The International Mouse Phenotyping Consortium. *Nat Genet.* 2017;49(8):1231-1238. doi:10.1038/ng.3901. PMID: 28650483

Overall Assessment

Typo issue:

In Supplementary Figure 5 (b–d), the cluster labels are not aligned with panel (a); panels (b–d) appear to be labeled 1–7, whereas panel (a) is labeled 0–6.

The same labeling inconsistency is present in Supplementary Figure 6 (b–d). The cluster labels should be corrected for consistency across panels.

Response: Completed as suggested.

Reviewer #3

1. While not raised in my initial review, I note that concerns about acknowledgment of previous results, raised by reviewer 1, remain. The authors should note that the change in beta-cell proportions in T2D have been noted previously. Even a short “similar to previous reports” within the results section would be sufficient. The authors state I their rebuttal that findings need to be repeated, which is true, but repetition must be acknowledged as such. This is a matter of scientific integrity.

Response: In Results, lines 116-120 of the R1 revised manuscript, we included this important acknowledgment in our phrasing and the relevant citations (one of which, PMID: 37231096, was highlighted again by Reviewer #1 in reviews of the revised manuscript).

However, we have further revised the manuscript text, lines 119-122, to move our acknowledgement of previous reports from the end of the sentence in R1 revised manuscript to the beginning in this R2 revised manuscript text:

“Consistent with the previous reports^{13,14}, we observed 13-15% lower overall β -cell/endocrine proportions in T2D islets than those in ND or PD donor islets (**Figure 1e**; mean β -cell percentages: T2D=42.2 \pm 11.3; ND=55.2 \pm 10.7, $p=0.006$; PD=57.2 \pm 12.9, $p=0.002$, ANOVA followed by Tukey's honest significance test).”

References Cited:

13. Wang, G. et al. Integrating genetics with single-cell multiomic measurements across disease states identifies mechanisms of beta cell dysfunction in type 2 diabetes. *Nat Genet* **55**, 984–994 (2023). PMID: 37231096

14. Wu, M. et al. Single-cell analysis of the human pancreas in type 2 diabetes using multi-spectral imaging mass cytometry. *Cell Rep* **37**, 109919 (2021). PMID: 34731614

2. The authors' data reveal more differential genes than other similar analyses, but the manuscript still lacks an explanation for these differences. In their response to reviewers, the authors indicated this was due to fewer contaminating non-endocrine cells, higher cell numbers, and greater sequencing depth. However, this critical information was not incorporated into the manuscript.

Response: The Results section has been revised to include quantitative features underlying the high quality of our dataset.

Revised Figure 1d now shows donor-specific cellular proportions of endocrine, exocrine, and immune cell fractions (left panel) and distribution of all cell types captured and identified (right panel), rather than endocrine-restricted proportions.

Lines 87-92: "This cohort yielded more high-quality cells per donor (1.73×, **Supplementary Figure 2a**), a higher proportion of β -cells (3.61×) and other endocrine cell types, and fewer contaminating acinar cells (0.37×) than a similarly sized cohort⁶. Additionally, deeper sequencing per cell type (**Supplementary Figure 2b**) delivered more expressed genes detected per cell type except proliferating α -cells (**Supplementary Figure 2c**)."

Reference Cited:

6. Elgamal, R. M. et al. An Integrated Map of Cell Type-Specific Gene Expression in Pancreatic Islets. *Diabetes* 72, 1719–1728 (2023) PMID: 37582230

Lines 187-190: "In contrast, we identified 746 β -cell DEGs at FDR<5% (T2D vs ND, **Supplementary Table 8**), ~10x those detected at FDR<10% in a recent HPAP cohort-based study⁶, likely resulting from greater sequencing depth and increased number of high-quality and endocrine cells captured in this cohort (**Supplementary Figure 2a-c**)."

Although we are hesitant to do so because it could offend or produce unwanted friction with HPAP investigators who generated the data or authors of the cited study who analyzed it, we have currently included the summary supporting these differences in **revised Supplementary Figure 2a-c**. At the reviewer's and/or editor's recommendation(s), we will be happy to remove this revised supplementary figure from the final manuscript.

As this is being presented as a resource paper, complete methodological transparency is essential. Future users need to understand not only what data are available, but how to generate comparable datasets. The technical improvements and comparisons to other datasets should be thoroughly documented in the manuscript, including:

- Specific protocol differences that reduced non-endocrine contamination
- Cell yield comparisons:
- Reads per cell per donor:
- Any differences in tissue handling or dissociation methods.

This information would strengthen the resource value by providing both improved data and improve methods for the community.

A thorough technical comparison would also provide valuable methodological insights that could inform future protocol optimizations, including potential updates to standardized procedures.

Response: As a Resource rather than Methods Development paper, comprehensive comparison of technical parameters is beyond the scope of this study. Additionally, because our research group does not harvest pancreas or isolate islets, we believe our evaluation is unlikely to yield objective, rather than subjective, insights into the quality of the procedures. We procure islets from islet isolation centers in the NIDDK-sponsored Integrated Islet Distribution Program or from ProdoLabs, who utilize established, standardized islet isolation and preparation protocols. Therefore, this study does not offer any insights or improvements over these gold standard protocols.

However, we have updated the Methods in the manuscript to better document our post-receipt islet recovery, dissociation, processing/monitoring steps as requested for better transparency and to enable investigators in the field to evaluate and compare approaches between studies.

- Specific protocol differences that reduced non-endocrine contamination:

Lines 621-623: “Only donor islet samples from IIDP or ProdoLabs with reported purity $\geq 80\%$ (median = 90%; range = 80-98%) and viability $>90\%$ were accepted for inclusion in this study.”

- Cell yield comparisons:

Lines 628-630: “Islet cell viability after dissociation was measured using a Countess automated cell counter (Invitrogen), with median post-dissociation viability = 77% (range = 59-89%; **Supplementary Table 1**).”

- Reads per cell per donor:

Lines 660-661: “Reads per cell per donor are included in **Supplementary Table 2**.”

- Any differences in tissue handling or dissociation methods:

Lines 623-630: “Upon receipt from IIDP or ProdoLabs, islets were recovered overnight in ProdoLab media following the ProdoLabs protocol (<https://prodolabs.com/protocols/>). Islets were dissociated into single cell suspension by incubating in Accutase at a ratio of 1,000 IEQ per ml for 8 minutes at 37°C with trituration every 2 minutes, as reported before¹⁰. Depending upon the reported islet index for each IIDP/ProdoLab isolation, we used 200-1000 IEQs per donor for dissociation into single-cell suspension. Islet cell viability after dissociation was measured using a Countess cell counter (Thermo Scientific) with Median post-dissociation viability = 77% (range = 59-89%; **Supplementary Table 1**).”

A thorough technical comparison would also provide valuable methodological insights that could inform future protocol optimizations, including potential updates to standardized procedures.

Revised Methods now includes updates and optimizations applied to the standardized protocols:

Lines 633-642: “Dissociated cells were washed twice and suspended in PBS containing 0.04% BSA, then filtered through a 40 µm Flowmi cell strainer to remove clumped cells and immediately processed as follows. Cell viability was assessed on a Countess automated cell counter (Invitrogen), and 12,000 cells from each suspension were loaded onto one lane of a 10x Genomics Chip G. Single cell capture, barcoding, and library preparation were performed using the 10X Chromium platform (<https://www.10xgenomics.com>) according to the manufacturer’s protocol for chemistries v2 (#CG00052) and v3 (#CG000183). cDNA and libraries were checked for quality using TapeStation 4200 (Agilent) and Qubit Fluorometer (ThermoFisher), quantified by KAPA qPCR, and sequenced on an Illumina NovaSeq 6000 (S1, S2 or S4 100 cycle flow cell), targeting 6,000 barcoded cells with an average sequencing depth of 50,000 reads per cell.

3. How many IEQs were used per donor? Please include the range per donor in the methods, since this may have contributed to the higher number of cells captured per donor.

Response: We loaded 12,000 cells per donor. To improve the reproducibility by other users we have included the range of IEQ used per donor in the method section:

Lines 626-628: “Depending upon the reported islet index for each IIDP/ProdoLab isolation, we used 200-1000 IEQs per donor for dissociation into single-cell suspension.”

4. Please remove UNOS ID from donors and include only RRIDs. UNOS ID are protected health information (PHI) and should be avoided since living relatives can still be affected by sequencing data. My apologies for not noting this problem on the prior review.

Response: We thank the reviewer for catching this important mistake. UNOS IDs were included for 6 of the 48 donors; *they have been removed from revised Supplementary Table 1* along with any affiliated metadata supporting deposited SRA or cell x gene profiles.

Minor points.

1. Lines 147-152 would benefit from revision for clarity. Consider breaking it into shorter sentences and ensuring consistent formatting of cluster assignments and cell types. **Completed as requested.**
2. Line 118. Please write out “percentages”. **Completed as requested.**
3. Line 137. Please remove “were”. **Completed as requested.**

Dear Dr Stitzel,

Thank you again for the submission of your amended study (EMBOJ-2026-123569-T) to The EMBO Journal. We have carefully assessed your revised manuscript, and the point-by-point response provided to the referee concerns that were raised before during review at a different journal. In addition, and as mentioned before, we decided to send the revised version of your work back to the original reviewer #3 for arbitrating reassessment with respect to technical robustness, and overall suitability of your work for publication in The EMBO Journal. We have now received his/her re-report which I enclose below. As you will see, this expert is now in favour of the work and supportive of publication at The EMBO Journal, pending satisfactory minor revision.

We are thus pleased to inform you that we can swiftly move forward towards acceptance of this work at The EMBO Journal, pending minor revision.

Please consider the remaining points by the referee carefully and adjust the discussion of the results in light of potential contributing parameters, introducing caveats and amendments where appropriate.

We also now need you to take care of a number of minor points related to formatting and data annotation, which I will share shortly in a separate message, together with additional changes and requests by our production team for Source Data provision.

Please submit a revised version of the manuscript using the link enclosed below, addressing the advisor's comments.

As you might have seen on our web page, every paper at the EMBO Journal now includes a 'Synopsis', displayed on the html and freely accessible to all readers. The synopsis includes a 'model' figure as well as 2-5 one-short-sentence bullet points that summarize the article. I would appreciate if you could provide this figure and the bullet points.

Please let me know any time should you have additional questions regarding above points.

Thank you again for giving us the chance to consider your manuscript for The EMBO Journal, I look forward to hearing from you and receiving your final revised version of the manuscript.

Best regards,

Daniel Klimmeck

Please remember: Digital image enhancement is acceptable practice, as long as it accurately represents the original data and conforms to community standards. If a figure has been subjected to significant electronic manipulation, this must be noted in the figure legend or in the 'Methods' section. The editors reserve the right to request original versions of figures and the original images that were used to assemble the figure.

Arbitrating re-comments, Referee #3:

This manuscript from Bandesh et al. provides analysis of control versus T2D and prediabetic human islet cells, identifies more DEGs than other similar studies, and includes some supporting mechanistic data. Bandesh et al. have improved their manuscript by adding experimental detail regarding islet treatment and comparisons between their islet cell parameters and those of the Elgamal et al. study. While the authors attribute differences in DEG detection to sample purity, cell number captured, and sequencing depth, they do not acknowledge that analytical differences, e.g. cell-type assignment, which appears differ in the number of markers used (one versus 2+), cut-off stringency ,and pipeline parameters, could also contribute. They have not tested their claims experimentally, so at minimum, this alternative explanation should be discussed.

Additionally, the authors' concern about offending other study authors is not a valid scientific justification for omitting these acknowledgments. Transparent discussion of methodological differences is standard scientific practice and does not imply criticism of other studies; rather, it reflects appropriate caution in interpreting cross-study comparisons. For example, noting the use of >80% islet purity may encourage people using other datasets such as HPAP to apply similar filtering criteria.

Dear Dr Stitzel,

Further to below, I attach the mentioned list of formatting changes required for your final resubmission.

Please let us know any time should you have additional questions related.

Best regards,

Daniel Klimmeck

>> Please add up to five keywords to your study.

>> Limit the abstract to maximally 175 words.

>> Adjust the title of the 'Conflict of Interests' section to 'Disclosure and Competing Interests Statement'

>> Please correct the order of the manuscript sections to: Abstract / Introduction / Results / Discussion / Methods / Data Availability / Acknowledgements / Disclosure and Competing Interests Statement / References / Figure Legends

>> References: please adjust reference format to EMBO Journal format, 10 authors et al.

>> Remove the 'data not shown' statements on p. 7 line 182, p. 8 line 203, or add respective information.

>> Please add a Reagents and Tools table to the Methods section, as a separate file using the existing template in the Guide For Authors, listing key reagents, experimental models, software and relevant equipment.

Please compare also our Guide-to-Authors instructions:

<https://www.embopress.org/page/journal/14602075/authorguide#structuredmethods>

>> Funding: please enter the following funding information in the list of funders in our online system: "American Diabetes Association grant 11-22-JDFPM-06".

>> Dataset EV legends: Rename Suppl. Table S1 - "Table EV1 - ". Rename Suppl. Tables S2 - 16 - "Dataset EV1" - "Dataset EV15". Please ensure that a legend is included in every file and that the headings are correct.

>> Please provide source data for the study as to the separate request e-mail. Source data should be uploaded as one (zipped) file per figure.

>>Appendix File with ToC: Please rename the file with the suppl. figures "Appendix", correct the figure s' nomenclature to "Appendix Figure S1" etc., and add a table of contents to the file, including page numbers.

>> Add complete annotation of human consent annotation to the Methods and adjust the Author Checklist sections Ethics and Data Availability accordingly.

>> Data availability section: please correct the header to "Data Availability".

>> Please indicate redisplay of data from Figure 3I & K in the legend of Figure 4F & H, respectively.

>> During our routine image checks, we noticed that the images across the figure set appear pixelated under analysis. This is a common result of converting original 16-bit TIFF images to RGB format for publication, and while not a cause for concern, it can sometimes give the impression of image alteration to critical readers.

To resolve this please upload the figure set at a higher resolution.

>> Consider additional changes and comments from our production team as indicated below:

- DAS:

Please note that the specific PRJNA913127, GSE221156 URLs for datasets are not provided in the data availability statement.

- Figure legends:

1. Please define the annotated p values ****/****/**/* as well as provide the exact p-values for the same in the legend of figure 5D as appropriate.
2. Please note that the exact p values are not provided in the legends of figures 2B-D
3. Please indicate the statistical test used for data analysis in the legends of figures 1B, E; 3C, E; 5B, C, D, F
4. Please note that the box plots need to be defined in terms of minima, maxima, centre, bounds of box and whiskers, and percentile in the legends of figures 3C
5. Please note that information related to n is missing in the legends of figures 1B, E; 3C, 5B, C, D, F, G
6. Please note that the error bars are not defined in the legends of figures 3D, 5F, G

Further information is available in our Guide For Authors: <https://link.springer.com/journal/44318/submission-guidelines>

Dear Dr Stitzel,

Thank you again for the submission of your amended study (EMBOJ-2026-123569-T) to The EMBO Journal. We have carefully assessed your revised manuscript, and the point-by-point response provided to the referee concerns that were raised before during review at a different journal. In addition, and as mentioned before, we decided to send the revised version of your work back to the original reviewer #3 for arbitrating reassessment with respect to technical robustness, and overall suitability of your work for publication in The EMBO Journal. We have now received his/her re-report which I enclose below. As you will see, this expert is now in favour of the work and supportive of publication at The EMBO Journal, pending satisfactory minor revision.

We are thus pleased to inform you that we can swiftly move forward towards acceptance of this work at The EMBO Journal, pending minor revision.

Please consider the remaining points by the referee carefully and adjust the discussion of the results in light of potential contributing parameters, introducing caveats and amendments where appropriate.

We also now need you to take care of a number of minor points related to formatting and data annotation, which I will share shortly in a separate message, together with additional changes and requests by our production team for Source Data provision.

Please submit a revised version of the manuscript using the link enclosed below, addressing the advisor's comments.

As you might have seen on our web page, every paper at the EMBO Journal now includes a 'Synopsis', displayed on the html and freely accessible to all readers. The synopsis includes a 'model' figure as well as 2-5 one-short-sentence bullet points that summarize the article. I would appreciate if you could provide this figure and the bullet points.

Please let me know any time should you have additional questions regarding above points.

Thank you again for giving us the chance to consider your manuscript for The EMBO Journal, I look forward to hearing from you and receiving your final revised version of the manuscript.

Best regards,

Daniel Klimmeck

Please remember: Digital image enhancement is acceptable practice, as long as it accurately represents the original data and conforms to community standards. If a figure has been subjected to significant electronic manipulation, this must be noted in the figure legend or in the 'Methods' section. The editors reserve the right to request original versions of figures and the original images that were used to assemble the figure.

Please use the link below to submit your revision:

Arbitrating re-comments, Referee #3:

This manuscript from Bandesh et al. provides analysis of control versus T2D and prediabetic human islet cells, identifies more DEGs than other similar studies, and includes some supporting mechanistic data.

Bandesh et al. have improved their manuscript by adding experimental detail regarding islet treatment and comparisons between their islet cell parameters and those of the Elgamal et al. study. While the authors attribute differences in DEG detection to sample purity, cell number captured, and sequencing depth, they do not acknowledge that analytical differences, e.g. cell-type assignment, which appears differ in the number of markers used (one versus 2+), cut-off stringency, and pipeline parameters, could also contribute. They have not tested their claims experimentally, so at minimum, this alternative explanation should be discussed.

Additionally, the authors' concern about offending other study authors is not a valid scientific justification for omitting these acknowledgments. Transparent discussion of methodological differences is standard scientific practice and does not imply criticism of other studies; rather, it reflects appropriate caution in interpreting cross-study comparisons. For example, noting the use of >80% islet purity may encourage people using other datasets such as HPAP to apply similar filtering criteria.

Arbitrating re-comments, Referee #3:

This manuscript from Bandesh et al. provides analysis of control versus T2D and prediabetic human islet cells, identifies more DEGs than other similar studies, and includes some supporting mechanistic data.

Bandesh et al. have improved their manuscript by adding experimental detail regarding islet treatment and comparisons between their islet cell parameters and those of the Elgamal et al. study. While the authors attribute differences in DEG detection to sample purity, cell number captured, and sequencing depth, they do not acknowledge that analytical differences, e.g. cell-type assignment, which appears differ in the number of markers used (one versus 2+), cut-off stringency, and pipeline parameters, could also contribute. They have not tested their claims experimentally, so at minimum, this alternative explanation should be discussed.

Response: As recommended, we now include a sentence acknowledging analytical differences as factors that could contribute to some of these differences in the revised manuscript, lines 194-196:

“... ~10x those detected at FDR<10% in a recent HPAP cohort-based study (Elgamal et al, 2023), likely resulting from greater sequencing depth and increased number of high-quality and endocrine cells captured in this cohort (Appendix Figures S2a-c. Alternatively, analytical differences, such as cut-off stringency and pipeline parameters, may contribute some of these differences. 511 beta-cell DEGs...”

Additionally, the authors' concern about offending other study authors is not a valid scientific justification for omitting these acknowledgments. Transparent discussion of methodological differences is standard scientific practice and does not imply criticism of other studies; rather, it reflects appropriate caution in interpreting cross-study comparisons. For example, noting the use of >80% islet purity may encourage people using other datasets such as HPAP to apply similar filtering criteria.

Response: We have retained the comparative components of the study in supplemental figures and tables as suggested by the reviewer.

Dear Dr Stitzel,

Thank you for submitting the revised version of your manuscript. I have now evaluated your amended manuscript and concluded that the remaining minor concerns have been sufficiently addressed.

I am thus pleased to inform you that your manuscript has been accepted for publication in the EMBO Journal.

Best regards,

Daniel Klimmeck

Daniel Klimmeck, PhD
Senior Editor
The EMBO Journal
EMBO
Postfach 1022-40
Meyerhofstrasse 1
D-69117 Heidelberg
contact@embojournal.org

Please note that it is The EMBO Journal policy for the transcript of the editorial process (containing referee reports and your response letters) to be published as an online supplement to each paper. If you should prefer removal of any referee-only figures included in the point-by-point response(s), e.g. because they may still be used for future publication or because they have been reproduced from published work by others, please do let us know immediately via response email.

More information is available here: <https://link.springer.com/partners/embo-press/editorial-policies#Peer%20review>